# Identifying and profiling structural similarities between Spike of SARS-CoV-2 and other viral or host proteins with Machaon

Panos Kakoulidis [1,2], Ioannis S. Vlachos[3,4,5,6,7], Dimitris Thanos[2], Gregory L. Blatch[8,9,10,11], Ioannis Z. Emiris[1,12] & Ema Anastasiadou [2✉]

Using protein structure to predict function, interactions, and evolutionary history is still an open challenge, with existing approaches relying extensively on protein homology and families. Here, we present Machaon, a data-driven method combining orientation invariant metrics on phi-psi angles, inter-residue contacts and surface complexity. It can be readily applied on whole structures or segments—such as domains and binding sites. Machaon was applied on SARS-CoV-2 Spike monomers of native, Delta and Omicron variants and identified correlations with a wide range of viral proteins from close to distant taxonomy ranks, as well as host proteins, such as ACE2 receptor. Machaon's meta-analysis of the results highlights structural, chemical and transcriptional similarities between the Spike monomer and human proteins, indicating a multi-level viral mimicry. This extended analysis also revealed relationships of the Spike protein with biological processes such as ubiquitination and angiogenesis and highlighted different patterns in virus attachment among the studied variants. Available at: https://machaonweb.com.

[1] Department of Informatics and Telecommunications, National and Kapodistrian University of Athens, Ilisia 157 84 Athens, Greece. [2] Biomedical Research Foundation of the Academy of Athens, 4 Soranou Ephessiou St., 115 27 Athens, Greece. [3] Broad Institute of MIT and Harvard, Merkin Building, 415 Main St., Cambridge, MA 02142, USA. [4] Cancer Research Institute, Beth Israel Deaconess Medical Center, 330 Brookline Avenue, Boston, MA 02215, USA. [5] Department of Pathology, Beth Israel Deaconess Medical Center, 330 Brookline Avenue, Boston, MA 02215, USA. [6] Harvard Medical School, 25 Shattuck Street, Boston, MA 02115, USA. [7] Spatial Technologies Unit, Harvard Medical School Initiative for RNA Medicine, Dana Building, Beth Israel Deaconess Medical Center, 330 Brookline Avenue, Boston, MA 02215, USA. [8] Biomedical Biotechnology Research Unit, Department of Biochemistry and Microbiology, Rhodes University, PO Box 94Makhanda (Grahamstown) 6140, Eastern Cape, South Africa. [9] Biomedical and Drug Discovery Research Group, Faculty of Health Sciences, Higher Colleges of Technology, PO 25026 Sharjah, UAE. [10] Institute for Health and Sport, Victoria University, Melbourne, PO Box 14428, VIC 8001 Melbourne, Australia. [11] The Vice Chancellery, The University of Notre Dame Australia, PO Box 1225, WA 6959 Fremantle, Australia. [12] ATHENA Research and Innovation Center, Artemidos 6 & Epidavrou 15125, Marousi, Greece. ✉email: anastasiadou@bioacademy.gr

Protein structure strongly dictates function, with minimal changes often dramatically affecting major properties[1], localization[2], binding partners[3], and structure stability[4]. Protein tertiary structure can be up to ten times more conserved than genomic sequence[5] and can reveal a greater genomic similarity. Currently, access to protein structures unprecedentedly increased due to advances in experimental and in silico approaches, such as refined Cryo-EM[6] and AlphaFold[7]. This progress stresses the need for improved methodologies that can efficiently identify structurally similar proteins beyond distinct domains or common families and origins. Such approaches could connect structure to molecular and cellular function and/or genomic information with structural and chemical properties. These methods can also be applied to assess structural mimicry mechanisms employed by viruses to control host systems[8] or to quickly functionalize proteins of newly discovered viruses or variants, supporting biomarker discovery, drug design or repurposing.

The currently available methods span a wide gamut of approaches, focusing on primary, secondary, or tertiary protein structure. Sequence-based search methods, such as BLAST[9], rely on high sequence similarity, which is often inadequate for revealing distant relationships. Established structure-based approaches, such as the Dali server[10] and Research Collaboratory for Structural Bioinformatics Protein Data Bank's (RCSB PDB)[11] search based on BioZernike[12], emphasize the high rate of structural similarity with fixed thresholds. From a performance perspective, most standardized metrics for structural comparison[13], such as the Template Modeling Score (TM-Score)[14], require prior superposition of whole structures, which is a computationally intensive task. It is becoming apparent from the above that there is an unmet need for new methods to perform structural comparisons.

Structure comparison by multiple criteria has been previously proposed, such as pyMCPSC[15]. However, to our knowledge, no existing method combines features such as torsion angles[16,17], residue distances[10,18,19] or representations like alpha shapes[20] to a unified framework. In this study, we present Machaon, a methodology that relies on hypothesis-free clustering and ranking proteins by concurrently calculating and utilizing comparative protein differences of torsional angles (B-phipsi), inter-residue distances (W-rdist), and surface complexity (T-alpha). These metrics are not linked with the length or the orientation of a structure, and alignments are utilized solely in constrained comparisons to prune the search space[21]. Machaon not only identifies structurally similar proteins to a reference protein but also performs meta-analysis, building a profile with extended comparisons and analysis of the results. The meta-analysis module assesses the genomic or transcriptomic sequence and protein 1D/2D/3D/chemical structural similarities. It also proposes evolutionary relationships, indicating functions of the reference protein, by highlighting areas of its secondary structure.

As a proof-of-concept study, we applied Machaon on the Spike protein monomer of Severe Acute Respiratory Syndrome Coronavirus 2 (SARS-CoV-2) in both its native form and its Delta and Omicron variants. SARS-CoV-2 virus greatly impacts the host respiratory system causing excessive inflammation response[22] and affects the cell compositions and biological pathways of vital organs[23]. It carries transmembrane trimeric spike glycoproteins (S protein) on its envelope, binding to the Angiotensin-Converting Enzyme 2 (ACE2) receptor on the cellular membrane, allowing cell entry[24]. Machaon enabled us to efficiently and accurately identify structurally similar proteins from different proteomes by comparing the monomer of the S protein with three large datasets: a viral PDB dataset, a human PDB dataset and a predicted human proteome by AlphaFold. We extended our comparisons with the viral dataset on the domain level for a more fine-grained search. We also combined the metrics with mixed representation alignments for an in-depth constrained search to find structurally similar protein segments to binding sites. The accompanied meta-analysis allowed us to form hypotheses on distant evolutionary paths, potential functional structural motifs and associations between functions or pathways and the reference protein. Finally, we were able to investigate structural differences between the native, Delta and Omicron Spike monomer variants by comparing Machaon's results for each monomer on the viral dataset.

## Results

**Overview of the method**. Machaon is an in silico analysis suite that identifies and selects the most structurally similar proteins from a user-specified pool of candidates, having a protein of interest as a reference (Fig. 1). This operation relies on torsion angles, residue distances and atomic coordinates from PDB data. It computes distribution distances of the multivariate phi-psi angle pair (B-phipsi), Wasserstein intra-molecular distances (W-rdist) and surface complexity difference (T-alpha), either for a whole structure or a part of it (see "Methods"). The three metrics are combined into vectors that represent the proteins in a 3D search space, which is pruned by keeping a set of top points for each metric. The method assigns each candidate protein to a cluster by its metric vector and chooses a cluster by considering a mixed ranking of the metrics. Machaon builds a profile of each finalist protein with metadata from established third-party services (Supplementary Fig. 1) and performs filtering by the coding gene. The method extends this profile by computing established metrics for tertiary and chemical structure similarity and by performing sequence alignments between the transcripts and protein secondary structures of each identified protein and the reference molecule. Finally, Machaon provides indications of the reference protein's functions based on the accumulated information of the selected proteins, pinpointing potentially related areas in the reference protein 2D structure.

**Validation of the method's accuracy**. To validate the method, we appraised the accuracy and limitations of Machaon on tasks of whole structure comparisons on different cases and datasets (Table 1). We retrieved the datasets from public repositories to avoid biased results. Machaon compares two proteins in an alignment-free manner and a varying granularity that extends to smaller/dispersed common fragments beyond matching domains. This property applies to cases where the protein does not have large common consecutive parts with the target dataset. Thus, our proposed method cannot be directly compared with methods focusing on structural alignment or being evaluated on restricted domain searches. The interpretation of our tests emphasizes the structural similarity of the identified proteins, which is an effective validation strategy, given the absence of a universal ground truth[18]. We measure 3D similarity with TM-Score[25], an established metric that is independent of protein length, and 2D sequence identity from 2D fold sequence alignments (see "Methods: Enrichment and assessment of the selected candidate entries"). For validation purposes, the results are pseudo-labeled based on the specific criteria of each task.

We initially investigated Machaon's detection scope on identifying proteins of the same CATH[26] (class, architecture, topology, homology) family utilizing BioZernike's validation set. This dataset is a non-redundant set of 2685 structures belonging to 151 CATH families. We employed Machaon for a random protein per family in the set (Fig. 2a) for families with more than

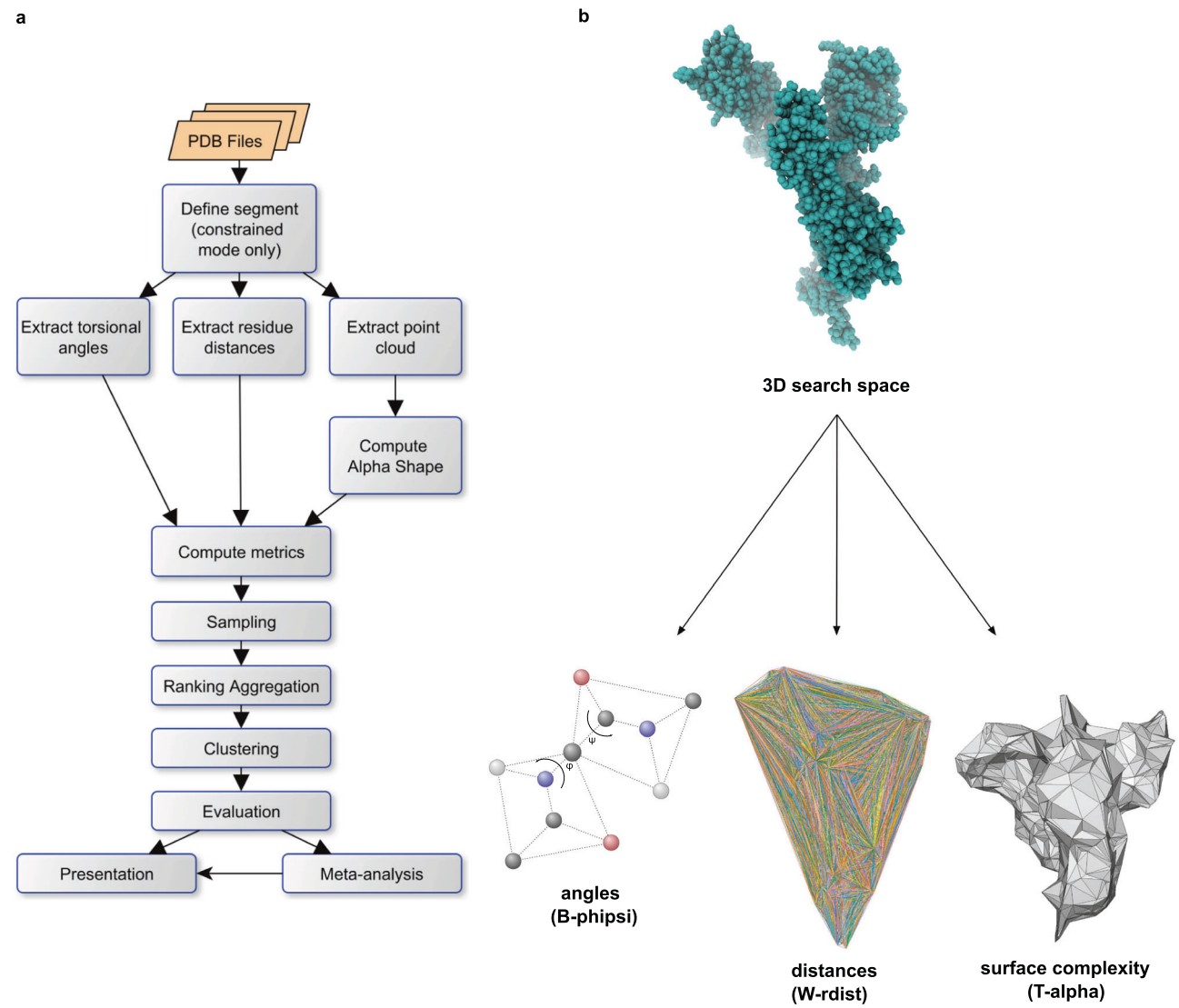

**Fig. 1 The workflow of Machaon and its perspective on the protein structure. a** A protein of choice is assigned as a reference for the comparisons with a selected dataset. The metrics are computed independently between the features extracted for each protein in the dataset and the reference ones. For segment comparisons, there is an additional initial step where the segments are defined based on mixed sequence alignments, which represent similarity in secondary structure and protein sequence-derived hydrophobicity level. The final set of candidate proteins is determined by ranking aggregation and clustering. There is an intermediate pruning step that samples the top 1% entries per metric order in case of a large dataset. The final cluster is enriched with Uniprot/Gene Ontology (GO) data and evaluated in protein, transcript and chemical levels. The method performs GO meta-analysis to reveal novel relationships and functions by aggregating the properties of the output set. **b** Machaon relaxes the structure alignment problem to a non-pairwise comparison of angles, distances and surface complexity through metrics that are not linked with the orientation of a structure; there is no requirement for prior alignment.

one occurrence. We labeled the final entries that belong to the CATH family of the reference protein as true positives. The method found the true positives with high recall/specificity, low precision and f1-scores close to zero since many more structurally relevant proteins were included in the final results (Table 1). We treated the latter entries as false positives for the sake of the task's measurements. The true and false positives had high and medium-to-low TM-Scores, respectively, which conveyed that Machaon operates outside the CATH system (Fig. 2b). A subset of proteins with different superfamilies/families in the final cluster exhibited higher structural similarity than some proteins of the target CATH hierarchy, which is in accordance with previously reported observations[21]. These identified proteins would have been overlooked if our method had a search criterion restricted to the CATH classification.

The second task involved the dataset of SHREC 2018 challenge[27] (2267 structures deriving from the conformational space of 107 proteins) and a series of comparisons between every protein represented more than once and the rest of the proteins in the set (Fig. 2c, d). The true positives in this task were the proteins in the results that were alternative conformations of the reference protein, as noted in the public data of the SHREC challenge. The method detected the majority of conformations with high recall/specificity, and the results were not limited to one protein including other similar structures in the set (Table 1). This resulted in a low F1 score and precision on the limited scope of the task. Whole structure comparisons had low accuracy in cases like a flexible short protein loop (9 residues, named '1982') as a reference that had conformations with low TM-Score (0.33). We employed constrained search to address this edge case by

**Table 1 The accuracy metrics for the three testing tasks of Machaon's assessment.**

|  | Accuracy metrics | TM-Score (%) | 2D-Identity (%) |
|---|---|---|---|
| Task 1 (CATH) | Recall: 0.73 | TP: 78.2 | TP: 60.4 |
|  | Specificity: 0.91 | FP: 30.4 | FP: 35.3 |
|  | $F_1$-Score: 0.05 | FN: 76.6 | FN: 56.5 |
| Task 2 (SHREC) | Recall: 1.0 | TP: 90.2 | TP: 86.4 |
|  | Specificity: 0.90 | FP: 28.8 | FP: 38.3 |
|  | $F_1$-Score: 0.14 | FN: 70.3 | FN: 86.4 |
| Task 3 (SCOP) | $F_{max}$ (Fold): 0.02 |  |  |
|  | $F_{max}$ (Family): 0.03 | P: 29.8 | P: 37.3 |
|  | $F_{max}$ (Superfamily): 0.19 |  |  |

The displayed values correspond to median of medians per search session for a task. The $F_{max}$ values of the Task 3 were calculated with SCOP140 benchmark's publicly available scripts. The pseudo labels TP (True Positive), FP (False Positive), FN (False Negative), P (Positive) are determined by the different criteria per each task: first task's true positives were the proteins that belong to a specific CATH family, second task's true positives were the proteins that were alternative conformations of a specific protein and the third task's true positives were the proteins that belong to a specific SCOP class.

setting the loop as a reference segment, resulting in the detection of 16 of the 19 conformations in the top 250 entries yielded by the method.

We extended our evaluation to a third task involving the non-redundant dataset PDB70 (~15,211 structures) of SCOP140 benchmark[28], which represents the classified proteins in extended Structural Classification of Proteins (SCOPe) 2.07 from a snapshot of the entire PDB (2018). The provided dataset consisted merely of ATOM records, so we kept just the first conformation in each file for compatibility (1867 PDB files were edited). We also excluded the d2zjq51/q105A domain that contained only the coordinates of alpha-carbons (no angle information available), updating the ground truth table. $F_{max}$ scores were computed with the benchmark's publicly available code for the untruncated final cluster per query domain, evaluating the ranking of the proteins in the results against the target SCOP classification (Table 1). The low scores suggested that Machaon identifies structurally similar proteins beyond SCOP folding classes (Fig. 2e, f).

**Identifying structurally similar proteins to SARS-CoV-2 Spike monomer.** The aforementioned tests provide evidence that the proposed technique was adaptive and fuzzy enough to be applied to a previously unknown viral protein for which no a priori information was available at the time. We applied Machaon to perform a large number of structural comparisons between Spike protein monomers of SARS-CoV-2 variants (native[29], Delta[30] and Omicron[31]) and a dataset of ~12,500 PDB files containing ~40,000 structures (median resolution: 2.64 Å) belonging mostly to proteomes of viruses (Fig. 3). We compared the proteins considering their whole available structure but also in a more localized scope such as domains or parts resembling the binding sites of the reference molecule. We also performed whole structural comparisons between the native Spike monomer and two datasets in separate sessions: a large dataset of ~160,000 structures, primarily consisting of human proteins (median resolution: 2.4 Å, Fig. 3) and a human proteome dataset as predicted by AlphaFold. Then, we investigated the properties of the Spike protein and its domains based on the emerging information by the viral dataset's comparisons.

As expected, whole structural comparisons between each of the three Spike variants (native, Delta, Omicron) and the viral dataset revealed several viral proteins belonging to the family of Coronaviridae in the top similarity results. However, viruses with great taxonomic distance, such as Human Immunodeficiency and Dengue[32,33] viruses from the Riboviria realm, the highest taxonomic rank of SARS-CoV-2 (Fig. 4a, Supplementary Figs. 2 and 3), also appear in the results. Moreover, Machaon detects proteins from viruses outside this viral realm, such as Herpesvirus, a virus that has been reported to be possibly reactivated by COVID-19[34]. Phage viruses belonging to the gut virome, such as Enterobacteria Phage T4, have a wide presence in the identified set as their expression is altered during COVID[35]. Non-viral proteins from host organisms ranging from bacteria to Homo Sapiens (Fig. 4b, c) also appear in the output set. We obtained entries of human proteins which have previously documented relationships with SARS-CoV-2, such as Complement (C3)[36], Dipeptidyl peptidase-4 (DDP4)[37] (related to Hypoxia) and Aminopeptidase N (ANPEP)[38,39] but also molecules without reported association such as Cleavage and Polyadenylation Specific Factor 1 (CPSF1), which is involved in a T cell pathway[40]. All mentioned proteins share structural similarities with Spike monomer in the 2D and 3D level as well as chemical and/or genomic/transcriptomic similarities—in 5'-end/3'-end Untranslated Regions (5'-UTR, 3'-UTR) and coding regions (CDS) (Supplementary Table 1, Fig. 5, Supplementary Figs. 4 and 5). These findings might indicate a potential viral mimicry of associated host proteins in genomic level[41], in addition to structural mimicry, forming a hypothesis for a multi-level manipulation of the host by the virus.

Following global comparisons, we conducted in-depth comparisons of domain and binding sites, using the native Spike monomer as a reference. We employed Machaon to identify proteins with structurally similar segments to the S1 N-terminal (NTD), S1 C-terminal (CTD), S1 receptor-binding (RBD) domains and to pre-computed binding sites, wherever it was applicable and possible by the available data. The results include a broader range of viral families, both distant and close, compared to those obtained from whole structure comparisons, like Papillomaviruses and Ebolaviruses (Supplementary Figs. 6–8). Additional non-viral proteins from eukaryotic organisms were encountered in the output sets, such as host receptors. On the binding site-targeted search, we observed that Spike protein has common structural elements with Angiotensin-converting enzyme 2 (ACE2), the known receptor of SARS-CoV-2 Spike protein (Fig. 6). Prior to their metrics-based ranking, these elements were aligned on mixed representations that encapsulate hydrophobicity and protein secondary structure information (5 levels of agreement in total). These associated areas belong to the predicted binding sites of Spike protein, and the corresponding correlated areas of ACE2 extended to its extracellular part. On a global scope, ACE2 PDB structure has 15.68% 3D similarity and 15.8% 2D sequence identity with the reference Spike protein PDB of native strain, meaning that they share common short structural areas that could include functional structural motifs. Additionally, the transcripts of ACE2 and Spike protein exhibit remarkable similarity (38.79% 5'-UTR, 37.72% CDS and 17.59% 3'-UTR sequence identities), supporting a potential correlation/co-regulation between Spike and ACE2 gene expressions.

The results on whole structural comparisons with two human protein datasets further highlight the mimetic nature of Spike's monomer structure toward the host proteins. The top results on the human experimental dataset enlist proteins related to the host's immune responses, such as Complement 5 (C5)[36], Apoptotic protease-activating factor 1[42] and Dicer1[43,44]. Proteins related to insulin[45], histones[46] and teneurins[47] are also included at high-ordered places in the final set. Regarding the results of the AlphaFoldDB dataset, macrophage mannose receptor 1

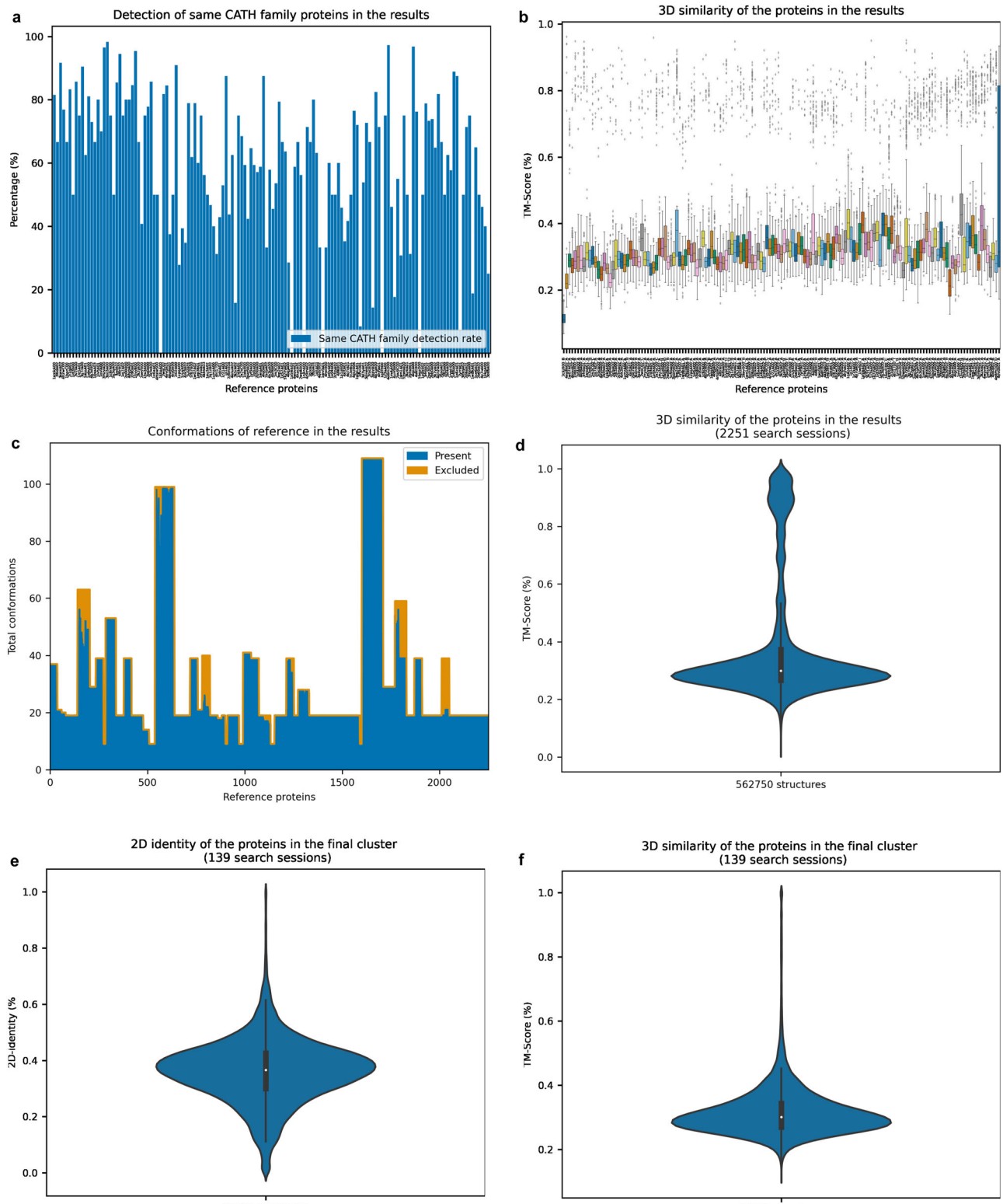

**Fig. 2 Results on CATH, SHREC and SCOP datasets, each of them used in a different testing task. a** A barplot depicting the detection rate of the proteins belonging to the same CATH family in the first task for each session of the reference proteins. Each CATH family is represented by a variable number of proteins in the dataset. **b** A boxplot per search session about the variability of 3D structure similarities of the proteins in the results (measured by TM-Align) on the first task. **c** An area plot that illustrates the amount of the conformations present in the second task's results for each referenced protein in the set (2251 search sessions total). **d** A violin plot that refers to the 3D structure similarity of the proteins in the second task's results as measured by TM-Align. The median TM-Score is above 0.17, which is the minimum value for a meaningful relationship between two structures. **e**, **f** Violin plots visualizing the 2D folds sequence identity and 3D similarity (TM-Score) for the proteins in the third task's results, respectively.

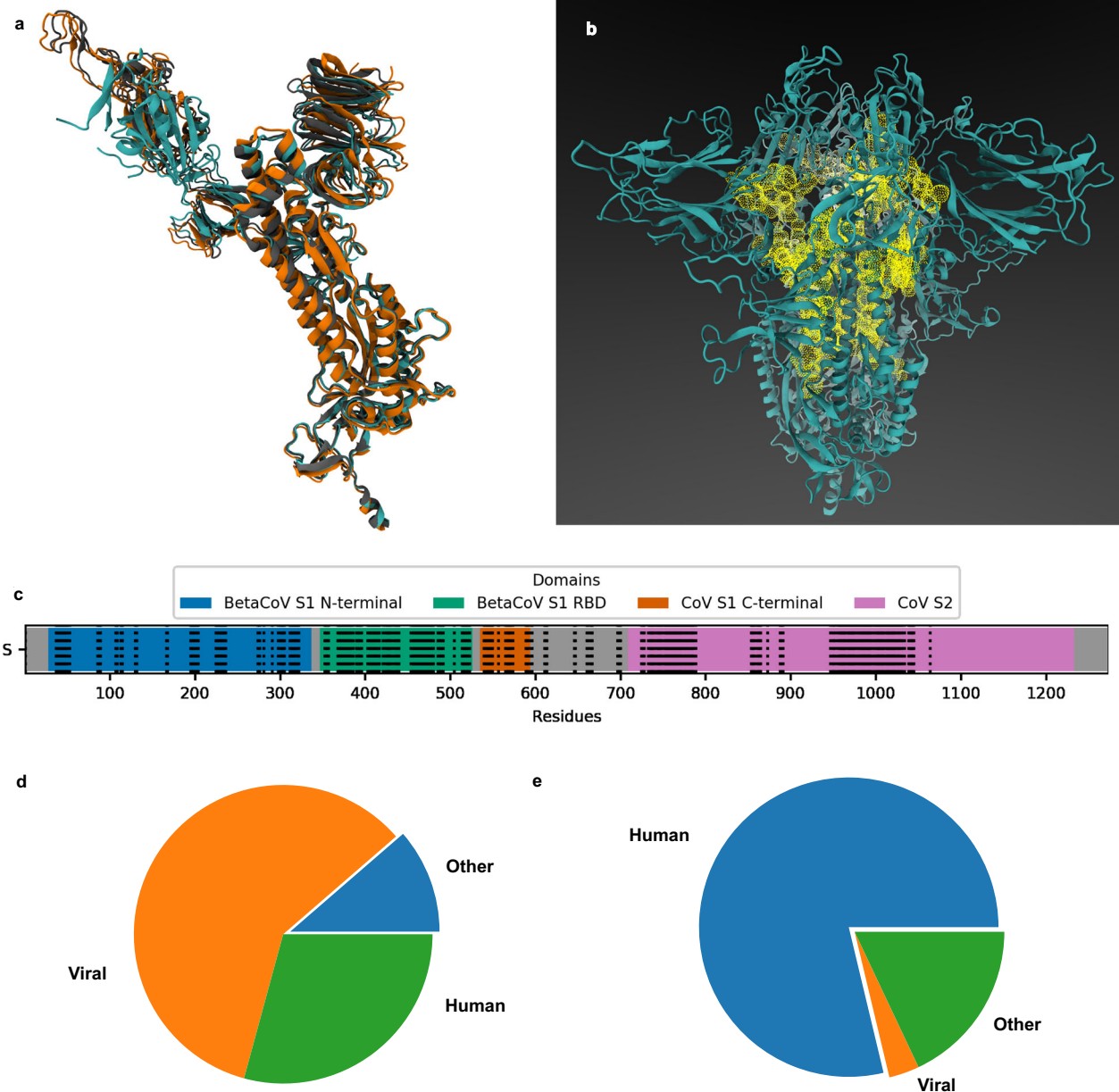

**Fig. 3 The data used for the study of SARS-CoV-2 Spike protein. a** The superimposed structures of Spike protein monomers of native (PDB IDs: 6VXX.A, color: cyan), Delta (PDB ID: 7V7Q.A, color: orange) and Omicron strains (PDB ID: 7T9K.A, color: gray) in Ribbons representation as shown in VMD. **b** The trimeric Spike protein structure is illustrated (closed state without glycans, PDB ID: 6VXX.A) in Ribbons representation as shown in VMD. The PDB data were prepared with Schrödinger Maestro, and the binding sites were estimated by the SiteMap module of the suite. The binding sites appear as yellow mesh spheres in the center of the bound Spike protein monomers. **c** The domains of SARS-CoV-2 Spike protein by residue position. The black dotted vertical lines represent the predicted binding sites. The gray area of the range 600–700 includes the cleavage site of the Spike protein. The total length of the protein is 1273 amino acids. **d**, **e** The compositions of the viral and human experimental datasets of PDB files. These were utilized as search spaces by Machaon, looking for structurally relevant proteins with SARS-CoV-2 Spike protein monomer in separate sessions. Each search space includes 'viral' as well as 'human' and 'other' sections, comprised, among others, of mammalian and bacterial proteins.

(MRC1)[48], Thyroglobulin (TG)[49], contactin-associated protein-like 3 (CTNNAP3)[50], Hypoxia up-regulated protein 1 (HYOU1)[51] are top entries with reported relationships to the virus. The method again ranks members of alpha-2-macroglobulin/complement 3 protein family (A2M/C3)[52] in high order: pregnancy zone protein (PZP)[53], CD109 antigen and Alpha-2-macroglobulin[54], which appears in the results of both human protein datasets. The selected candidates from these datasets share similarities in protein 2D/3D/chemical structure and transcript sequence, as observed in the results of the viral dataset

(Supplementary Figs. 9 and 10). Aggregating the connected Gene Ontology (GO) terms to the finalist proteins from both searching sessions on the human datasets, we noticed that most proteins were reported to reside in the nucleus, cellular membrane or extracellular space. Top GO terms were the ones that involve transcription with RNA polymerase II, cellular processes such as cell adhesion and functions of binding to metal ion, ATP and RNA. These findings enrich the indications of a multi-level and widespread viral mimicry for the modulation of the host system by SARS-CoV-2.

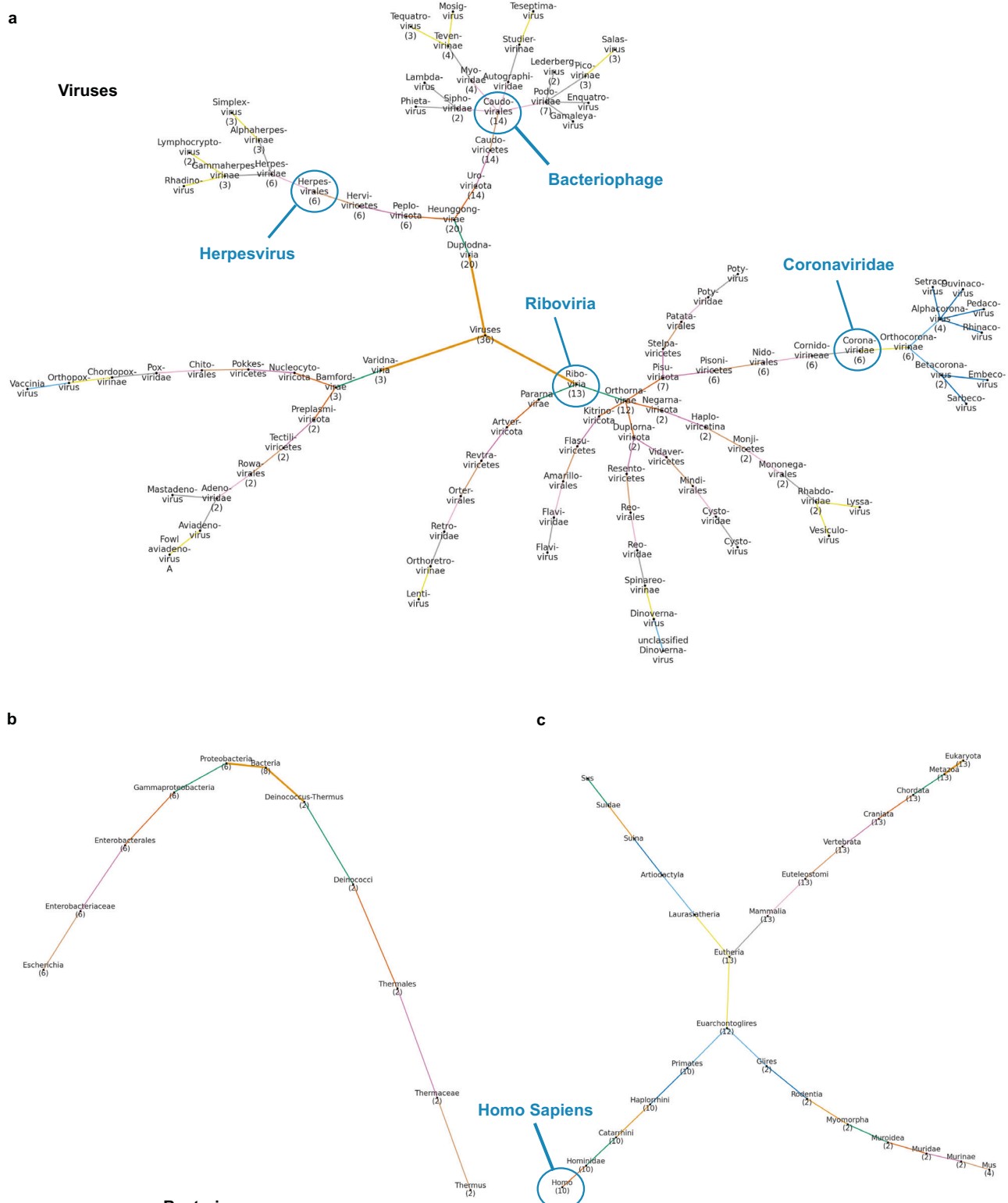

**Fig. 4 The lineage trees of the proteins in the final set of the whole structural comparisons of Spike (native strain) with the viral dataset.** These trees are generated by Machaon's presentation module. Each family name also carries a population number if there is more than one protein categorized under it. The root of the tree starts with thicker branches, and the colors designate the branch levels. The tree in (**a**) refers to the viral proteins that were found most structurally relevant, and the ones in (**b**, **c**) are generated for the correlated eukaryotic and bacterial proteins. These trees could also be treated as distant evolutionary trees according to the structural traits of the proteins in the results. The lineage information is retrieved from UniProt.

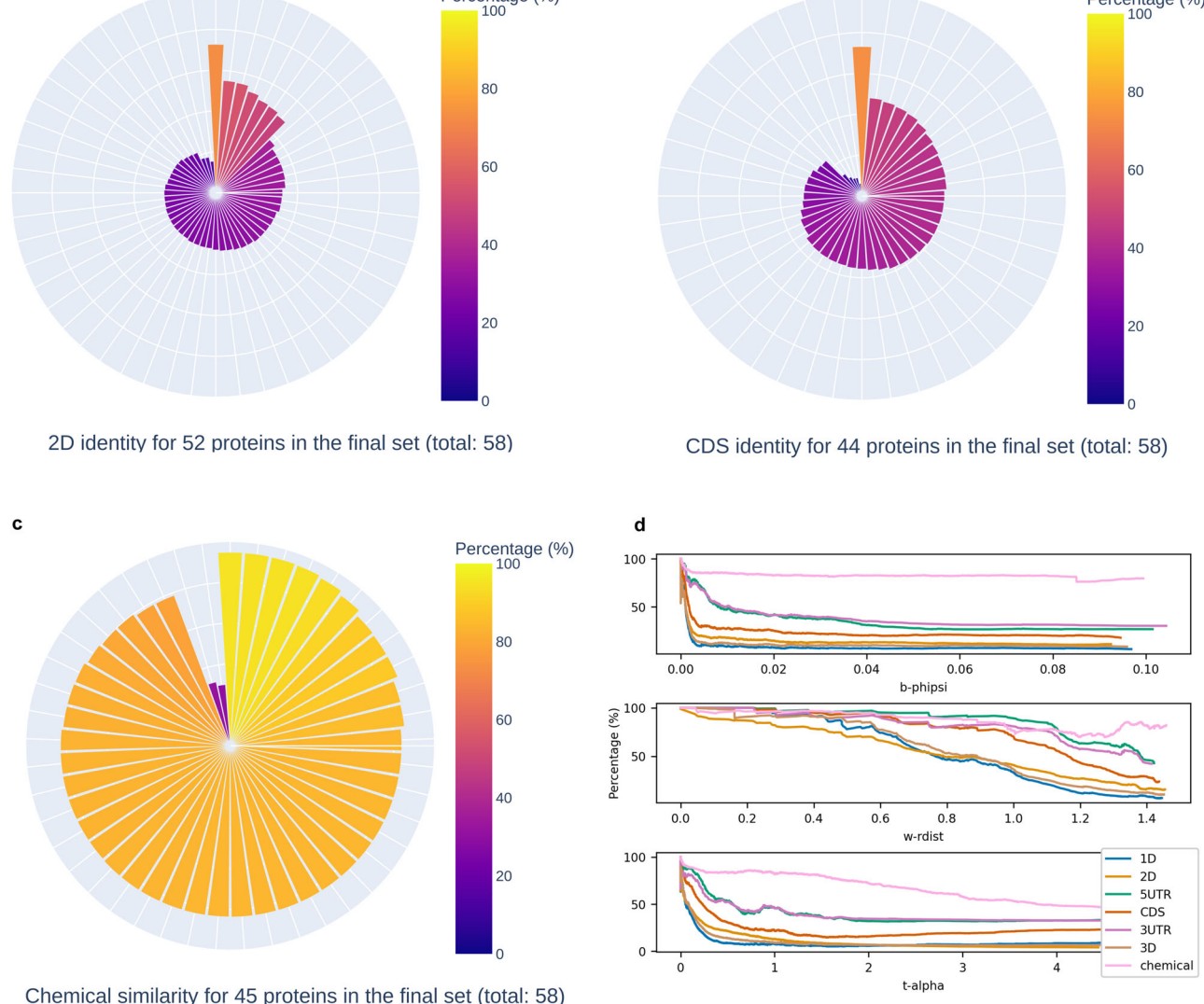

**Fig. 5 Interpretation of whole structure comparison results for Spike (native strain) with the viral dataset. a–c** Each protruding bar from the center of each radial plot represents a protein entry from the final set yielded by Machaon for whole structure comparisons between Spike (native strain) and the dataset. The plots are generated by the presentation module. The measurements were carried out by the evaluation module wherever the underlying data allowed it (missing or malformed relevant data). 2D folds sequence identities are depicted in (**a**), gene coding region identities in (**b**) and chemical similarities (Tanimoto Index) in (**c**). **d** This visualization is based on the comparison data between SARS-CoV-2 S protein (monomer, native strain) and separate sorted subsets for each metric (~7000 structures), sampled by the dataset (row position mod 5 of each metric's processed entries list). The figure illustrates the inverse relationship between the metric values and the identity/similarity percentages. Sequence identities for protein sequence retrieved from UniProt (1D), protein secondary structure estimated by PDB (2D), 5′-end Untranslated Region (5UTR), coding (CDS) and 3′-end Untranslated Region (3UTR) sequences from NCBI RefSeq, protein tertiary structure similarity computed by a modified TM-Align version from PDB (3D), and chemical similarities estimated from PDB. The identities were computed on global sequence alignments which were conducted with BioPython. All alignments followed the scoring of matches = 5, mismatches = −4, open gap penalty = −10, gap extension penalty = −0.5, except for the 1D alignments where matches/mismatches scores were based on the BLOSUM62 matrix. The values plotted are the EWMAs (span = 500) of the comparison metrics, which were pruned by absolute $z$-score (zscore < 0).

**Investigating the biological properties of the SARS-CoV-2 Spike protein.** Gathering information on protein structure similarities, in combination with meta-analysis, can unravel hidden molecular relationships and provide meaningful hypotheses for cellular functions. The meta-analysis module can aggregate proteins in the search results by a common Gene Ontology property and localize it on the protein secondary structure (Figs. 7 and 8). From the meta-analysis of viral dataset comparisons, we identified that the three versions of the Spike monomers (native, Delta, Omicron) have common structural elements with proteins related to ubiquitination[55,56], a process that has been

reported to be vital for virus replication[57] (Fig. 7a–c, Supplementary Table 2). This is in accordance with the findings from the comparisons of Spike monomer with the predicted human dataset as structural similarities with E3 ligases were unveiled, further enhancing the Spike protein's relationship with ubiquitination. We also performed whole structure comparisons with customized versions of the PDB files corresponding to the three variants of the Spike proteins (native, Delta, Omicron) to identify potential binding differences (Supplementary Data 2). These PDB files contain data on the intersection of the residue positions present in the three original PDB files and therefore each one

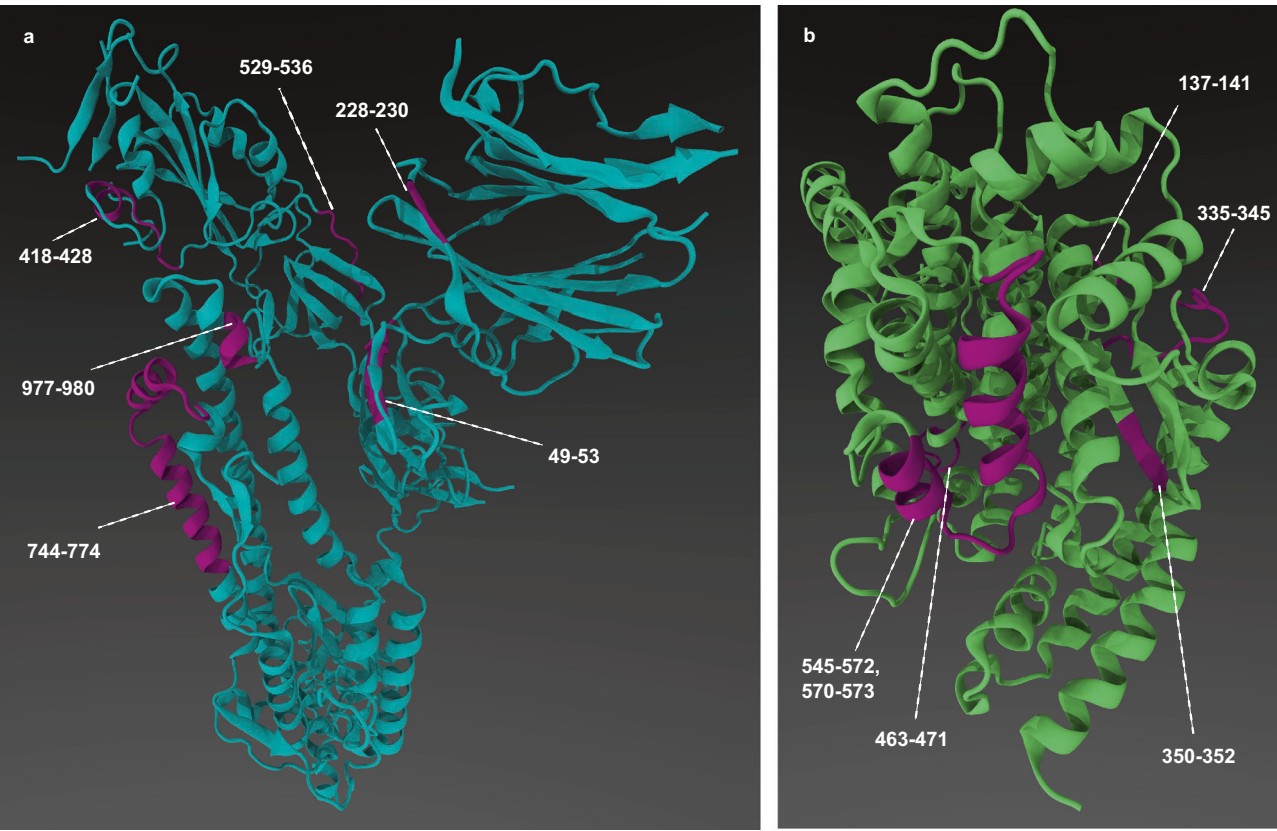

**Fig. 6 The associated fragments of native SARS CoV-2 S and ACE2 host receptor protein monomers.** Segments **a** in SARS-CoV-2 Spike protein (49–53, 226–230, 418–428, 529–536, 744–774, 977–980 residues, purple-colored areas, PDB 6VXX.A) were found to share structural and chemical similarities with segments **b** of ACE2 receptor (137–141, 335–345, 350–352, 463–471 and 545–573 residues, purple colored areas, PDB 3D0H.B). These areas are parts of a predicted Spike protein binding site and could potentially be parts of functional structural motifs common to both proteins. This information is derived from the constrained search of Machaon targeting an area where a Spike protein's binding site is associated with the ACE2 receptor via a mixed sequence alignment.

covers the same residue positions. Meta-analysis of the results concerning the virus attachment process conveys an altered binding pattern between the three proteins that could relate to the different infection rates (Fig. 7d–f). Additionally, this difference can be attributed to structural differences between the proteins, as we observed by comparing their secondary structures (Supplementary Fig. 11). Meta-analysis on the results of the NTD, CTD and RBD domains of the native Spike protein demonstrated links with host biological processes: angiogenesis[58] and CTD, blood coagulation[59] and RBD, heart-related[60] processes and NTD (Fig. 8). Similar secondary structural elements were identified with short protein domains in the dataset that could potentially pose as structural equivalents to short linear motifs (SLiMs)[61]. Outputs of whole and constrained comparisons are available in Supplementary Data 3 and 4 for the viral dataset and Supplementary Data 5 and 6 for whole structure comparisons with human datasets.

**Seeking biases in the results for SARS-CoV-2 Spike**. We tested how different setups would alter the previously presented results on Spike protein by showcasing the robustness of the metrics. First, we computed the metrics on different versions of SARS-CoV-2 Spike protein: alternative resolutions, conformations, lengths and variations (516 PDB chains). The measurements suggested that most of these versions would have been identified if they were present as unknown proteins in the whole structure comparisons results for Spike and viral dataset. The median

values of the metric vector [*B-phipsi, W-rdist, T-alpha*] in the final cluster were [0.0180, 0.9410, 0.0378] and for the Spike proteins set were [0.0025, 0.7334, 0.0666] (Supplementary Data 7). We also performed an alternative search session with Machaon having a preprocessed version of Spike's protein PDB structure (see "Methods: Prediction of Spike protein's binding sites") as a reference. This version extends to 102 more amino acids, and its refinements affect the conformation of the structure. Although preprocessing crystallographic data is a common procedure of their analysis protocol, it is an overhead step that requires high expertise and a toolchain of adequate software tools. PDB preprocessing is not critical for Machaon's performance since the intersection of the identified protein sets between the searches targeting the two PDB versions (raw and preprocessed) included 72% of the superset (Supplementary Data 8). The search with the raw version was fuzzier due to the shorter length of the reference, yielding a bigger unique subset mostly of phage, host and bacterial proteins, a pattern that was also observed in the unique subset for the preprocessed version.

We further assessed the results on whole structure comparisons with the viral dataset to interpret the metrics in the context of biology. Based on the correlation matrix (Supplementary Table 3), the metrics segregate on the same input 3D structures so their information does not overlap. Therefore, each metric is well-separated and targets a different aspect of the protein structure, so there are no redundant computations (metric space: Supplementary Fig. 12). Judging from the computations carried out by the evaluation module, the combined representation by the metrics

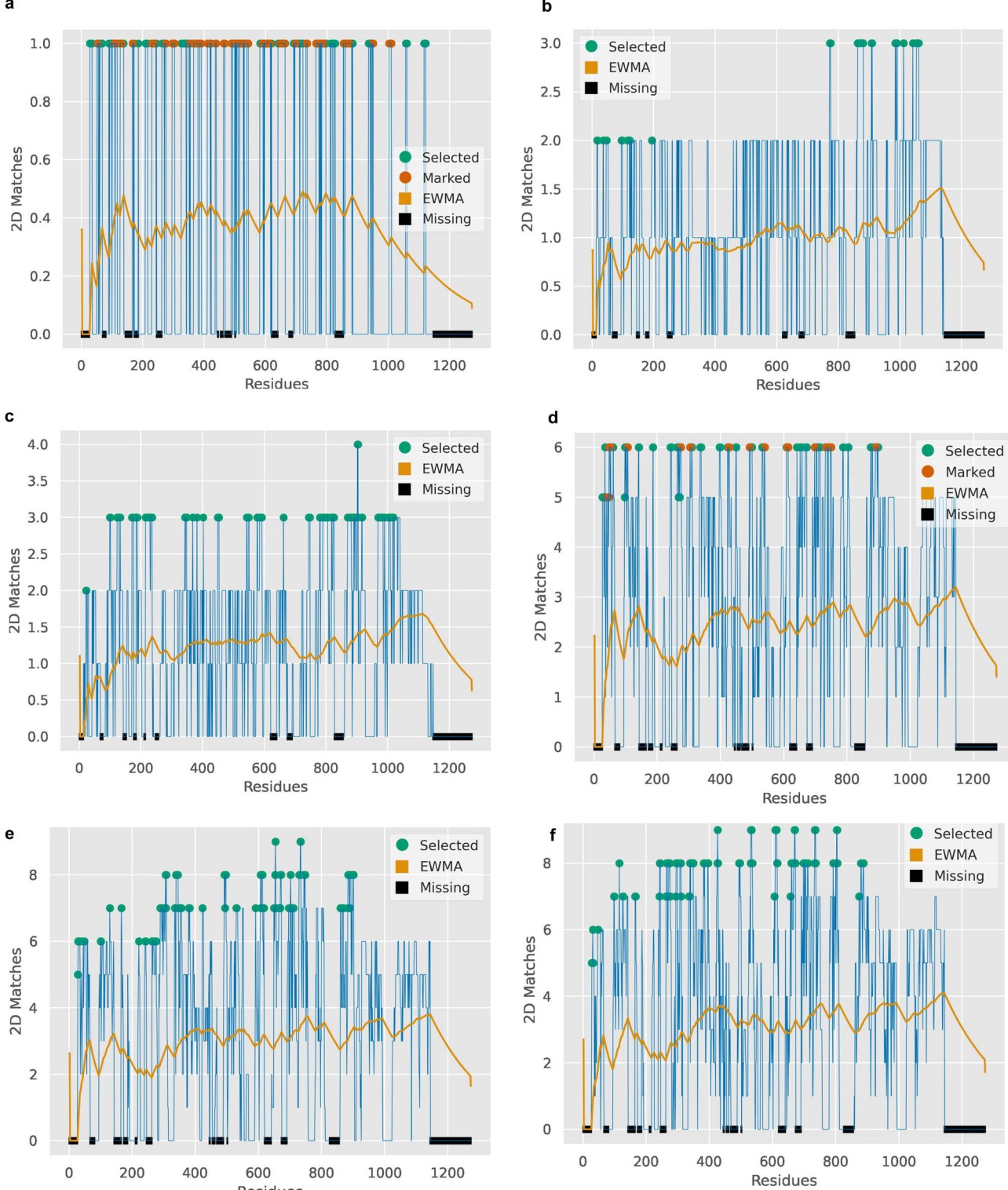

**Fig. 7 The meta-analysis module's visualization output for the three Spike proteins (native, Delta, Omicron) and ubiquitination, virus attachment processes.** These plots illustrate multiple protein secondary structure alignments of proteins in the results for the viral dataset. These proteins are connected with a Gene Ontology (GO) term in question. Peaks that are annotated with colored thick dots represent the residue positions that are associated the most with the target term. The ones with orange dots are characterized as positions that participate in binding sites (as calculated by SiteMap). The area where the exponentially weighted moving average (EWMA, yellow line, span: 30% of the reference sequence length) increases provides an indication of possible connections with the target property. Areas with missing residues in the reference PDB file are annotated with black boxes on the *x*-axis. **a**–**c** Plots are generated for the identified proteins with GO properties matching to the 'ubiquit' search term, in whole structure comparisons, with **a** native, **b** Delta or **c** Omicron Spike proteins as reference; **d**–**f** refer to the identified proteins in whole structure comparisons, with native (**d**), Delta (**e**) or Omicron (**f**) Spike proteins as reference, that are associated with 'receptor-mediated virion attachment to host cell' and/or 'virion attachment to host cell' GO terms. On (**d**–**f**), reference PDB files contain data on the same residue positions as a means toward a direct comparison of the Spike variants.

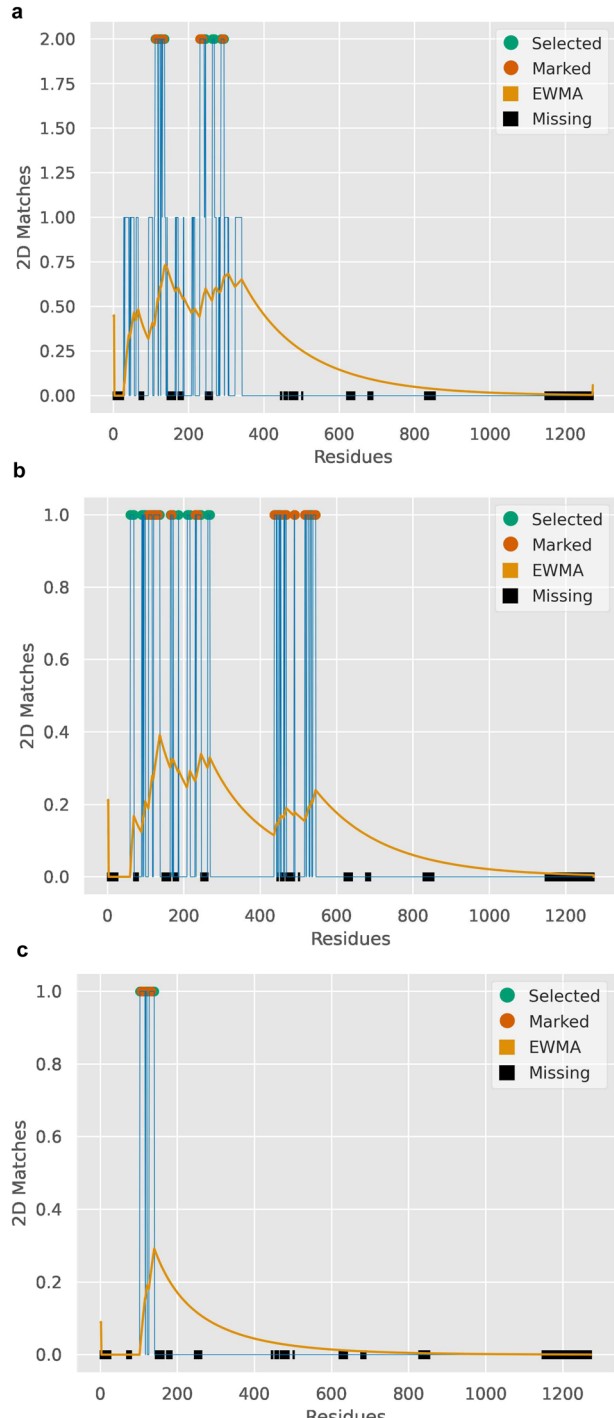

**Fig. 8 Meta-analysis for the results on the viral dataset and native Spike protein's CTD, NTD and RBD domains.** Positions with dotted peaks are selected by Machaon as the ones that are associated the most with the property in question. Selected peaks that fall into positions marked by the user are orange dotted, while green dotted peaks are uncharacterized. Here the marked sites are predicted binding sites of native Spike protein by SiteMap. The yellow line is the exponentially weighted moving average (EWMA, span: 30% of the reference sequence length) which visualizes the aggregated mean intensity of the positions' relation to the searched GO term. Areas with missing data in the reference PDB file are annotated with black boxes on the x-axis. These global alignments are on the protein secondary structure of the proteins that populate the final set and link with GO terms that match a search term. **a** is for the CTD domain and 'angiogen' search term, **b** is for the NTD domain and 'heart' term, and **c** is for the RBD domain and 'coagulation' term. We see that the S1 domain contains short structural elements that are similar to host proteins related to angiogenesis, heart process/development and blood coagulation.

a specific pattern on a 2D folding type, e.g., alpha helices, but the matches are spread out (Fig. 9) and follow the content of the Spike monomer (Supplementary Figs. 13–15) including loops, alpha helices and beta-sheets. Therefore, the proposed method is not biased toward a specific secondary structure motif.

## Discussion

Machaon is a methodology that identifies structurally similar proteins to a reference protein but also performs meta-analysis by extended comparisons and examination of the results. The novelty of our approach is that it regards each protein as an assembly of flexible moving parts and not a rigid hollow solid structure. It considers the arrangement of the protein's inner components (B-phipsi, W-rdist) and how this is reflected in the resulting surface (T-alpha). Machaon perceives angles and residue distances as unstructured data and emphasizes the shape of their respective distributions. B-phipsi and W-rdist metrics quantify the structural similarity of the two compared proteins according to the same occurrence of phi-psi angle pair or inter-residue values in their peptidic chains. T-alpha serves as a geometrical filter and efficiently quantifies surface complexity in a smoothened unit that is decoupled in size. The feature selection leans toward the side of the secondary protein structure, and this is verified by the results. Each of the features has the convenient property of being human-interpretable and carries meaningful information. The method attempts to reduce the structures in dimension, complexity and proportion, bringing them to the same scale and smoothing out minor differences (e.g., data irregularities). To our knowledge, Machaon is the first structural comparison methodology that combines these three features under two different settings: whole structure and mixed-alignment-informed segment scanning. This is linked with meta-analysis that can offer hints for biological properties via multiple secondary structure alignments. This module also interfaces with various well-established data sources and methods, enriching further the resulting output with extended comparisons to genomic, proteomic and chemical levels.

The results are derived from the structures available in the PDB file. Thus, it is possible that some protein structures might be incomplete and partially represented, a common fundamental obstacle found in these datasets. Data preparation is a labor-intensive and resource-intensive process that could insert biases. Through Machaon, many of the above-mentioned limitations are overcome due to the fundamental principles of the methodology. More specifically, Machaon analyzes the experimental data in their raw form without any preprocessing or refinement, such as treating missing residues. Also, each protein may participate with

commonly associates with protein secondary structure and chemical similarity (Fig. 5). The 3D similarity of the selected proteins is lower than the 2D (Supplementary Table 1), showcasing the effectiveness of the method in a strict setting of searching distant structures. These structures would have been ignored, missing potentially important results if a conventional search was applied with a high minimum structural setting, e.g., >80%. We confirm this difference by conducting searches with alternative existing methods operating on protein sequence[9] or structure[10,11,26,62,63] (Supplementary Table 4). However, a direct comparison with Machaon is not possible since these methods differ in scope and approach. Also, the 2D pairwise alignments between the Spike monomer and all finalist proteins do not form

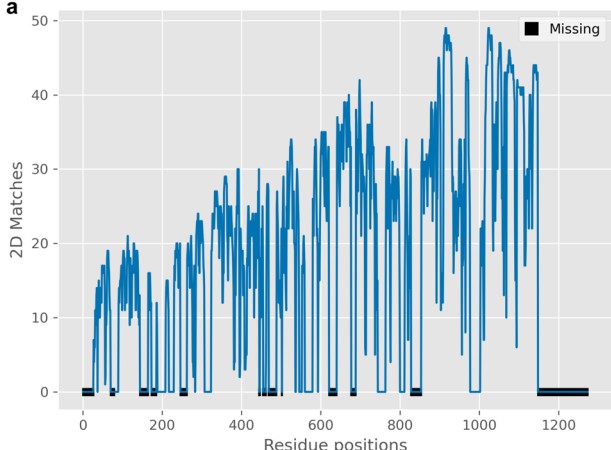

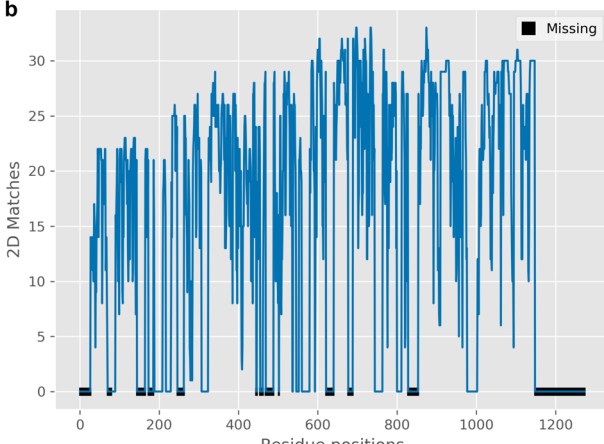

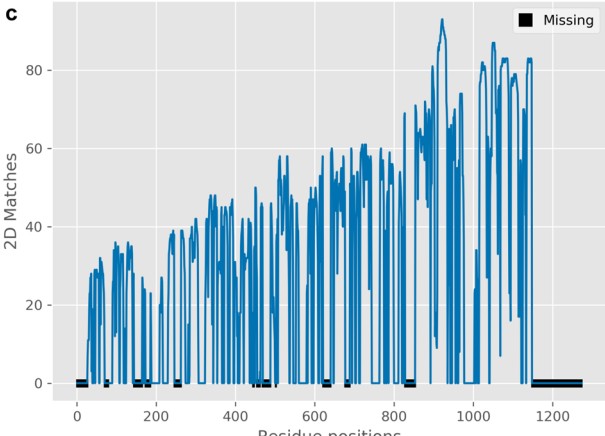

**Fig. 9 Plots on the aggregated 2D folding sequence alignments between the Spike protein monomer (native) and the final sets of whole structure comparisons.** 2D sequences were determined with DSSP/STRIDE and the matches derive from global pairwise alignments between each of the proteins in each dataset and the reference protein, the Spike monomer. **a** depicts the matches for the viral dataset, **b** refers to the comparisons with the experimental human dataset and **c** with the AlphaFold's human proteome dataset. Areas with missing residues in the reference PDB file are annotated with black boxes on the x-axis.

several structures in the dataset due to the incompleteness of the available data, as two different PDBs of the same protein might contain different parts of the structure in different conformations and resolutions. This is highly desired since the ideal dataset for

this kind of task would comprise information on different resolutions, conformations, protonation states, isoforms, etc. The metrics appear to be robust against a degree of noise, as demonstrated by the high overlap observed between the results for the raw and preprocessed version of the Spike monomer.

We thoroughly tested Machaon on public datasets and three individual tasks. The median TM-Score for whole structure comparisons in all tasks was well above 0.17, accounting for a meaningful relationship between a pair of compared structures[25]. The presented results confirm that the method operates beyond domains or folding classes and retrieves similar structures in a wider scope, balancing between secondary and tertiary structure similarities. Machaon's constrained searches offer fine-grained scanning with a manageable performance cost, identifying alternative conformations of the same protein that have very low TM-Scores. This particular case attests to the difficulty of structural search since there is no perfect solution or indisputable ground truth. An alternative conformation of the same protein having a TM-Score lower than 0.5 would be overlooked in favor of a protein from the nearest CATH or SCOP class[64]. Machaon is not relying on prior knowledge or hard cutoffs and does not discriminate candidates during scanning. Its data-driven manner, combined with its fuzzy metrics, is suitable for the investigation of challenging cases such as the novel Spike protein of SARS-CoV-2.

Identifying proteins with common folds could reveal distant evolutionary relationships or common traits from convergence evolution since structure is more conserved than sequence. This, in turn, provides further justification or indication about existing or novel common functions, cellular locations, protein interactions, structural motifs and pathways. The knowledge of structurally similar proteins could form hypotheses for intrinsically disordered proteins (IDPs) that belong to the dark proteome[65] and lack a defined 3D structure. The power and interest in such approaches became obvious with the appearance of the Alpha-Fold model[7]. The latter models distributions of dihedral angles and paired residue distances to predict the secondary structure, an intermediate step for the prediction of the tertiary structure. Our proposed method could be used with predicted data, such as the Human Proteome data from AlphaFold DB[66] or the meta-genomic proteins from ESM Metagenomics Atlas[67] and decipher previously unknown relationships. Machaon's metric values extend to the set of real positive numbers and close to zero, which are attractive traits for their potential adoption as objective functions in Machine Learning tasks. The suggested metrics could also be included ad hoc into Virtual Screening pipelines or provide known targets in drug repurposing studies by finding similar sites belonging to proteins that have been previously validated using ligand-binding studies.

We used Machaon to identify proteins that have structural similarities with SARS-CoV-2 Spike monomer. The results on the viral dataset further validate the accuracy of the method since many proteins with a high order belong to the Coronaviridae family. The generated viral taxonomy trees could initiate a further investigation of SARS-CoV-2's distant evolutionary paths. Judging from the lower median values of the gap included identities concerning the domain scanning, it is most possible that some of the identified domain similarities would have been missed in a global sequence or a maximum structural alignment context. The method ignores any sequence or structural gaps and considers similarity in a manner of local feature alignment. Therefore, it is suitable for finding less obvious, intermediate structural relationships, probably revealing valuable genomic similarity information or potential functional motifs, as in the case of the segment-scanning between Spike monomer and ACE2. Furthermore, the extended comparisons on sequence, structural and

chemical levels reveal more areas of relevance. The results for the viral and human datasets demonstrate such multi-level similarities between the Spike monomer and human proteins. This could hint at linked expression patterns with host genes, sharing or hijacking[41,44,68–70] regulatory relationships like transcription factors, miRNAs and immune system evasion or regulatory mechanisms[62,71–73]. For instance, it was recently reported that the SARS-CoV-2 virus hijacks host processes related to ubiquitination and that Spike protein participates in such a process[74]. These findings are in accordance with our indications of a potential Spike-ubiquitination relationship which could be researched further as a possible case of antagonism in the immune system's responses[71].

In brief, we investigated three structural metrics and combined them to establish a comparative analysis method. Machaon was shown to effectively encompass the suggested measurements in the context of global and segmented comparisons seeking structurally similar proteins. The proposed method was applied to investigate the SARS-CoV-2 Spike protein uncovering prominent similarities with different sets, including viral and host proteins. Some of the revealed proteins had previously documented relationships, further validating this technique. However, there were also novel findings involving the interactions or pathways of the proteins in the host organism that require further investigation.

Conclusively, we demonstrated that Machaon is a powerful approach that operates beyond conventional structural comparative methods. As structural comparison appears to provide more information than linear sequence comparison, it will be interesting to extend Machaon to other molecular structures apart from proteins, including DNA and RNA. Using an innovative comparative structure method, in addition to the conventional primary sequence alignments and an advanced meta-analysis tool, will provide a fresh angle of view and a powerful new approach for molecular studies.

## Methods

**Mathematical definitions.** Bhattacharyya distance[75] of phi-psi angles (b-phipsi) is a metric based on the multivariate distribution distance $D_B$ between two sets of phi-psi angle pairs $PP_A, PP_B$:

$$b_{phipsi}(PP_A, PP_B) = \frac{1}{8}\left(\boldsymbol{\mu}_{PP_A} - \boldsymbol{\mu}_{PP_B}\right)^T \boldsymbol{\Sigma}^{-1}\left(\boldsymbol{\mu}_{PP_A} - \boldsymbol{\mu}_{PP_B}\right) + \frac{1}{2}\ln\left(\frac{\det\boldsymbol{\Sigma}}{\sqrt{\det\boldsymbol{\Sigma}_{PP_A}\det\boldsymbol{\Sigma}_{PP_B}}}\right)$$

(1)

where $\boldsymbol{\mu}_{PP_A}$, $\boldsymbol{\mu}_{PP_B}$ and $\boldsymbol{\Sigma}_{PP_A}$, $\boldsymbol{\Sigma}_{PP_B}$ are the parameters of the corresponding distributions of $PP_A, PP_B$. $\boldsymbol{\Sigma}$ comes from the following equation:

$$\boldsymbol{\Sigma} = \frac{\boldsymbol{\Sigma}_{PP_A} + \boldsymbol{\Sigma}_{PP_B}}{2}$$

Wasserstein inter-residue contacts distance (w-rdist) focuses on intra-molecular paired residue distances. This metric is the log-normalized Wasserstein (or Earth's Mover) distance[76] between two distributions $d_A, d_B$ of inter-residue distances:

$$w_{rdist}(d_A, d_B) = \log_{10}\left(\left(\int_{-\infty}^{+\infty}|D_A - D_B|\right) + 1\right)$$

(2)

where $D_A, D_B$ are the cumulative distributions of $d_A, d_B$.

Only the $C_\alpha$ atoms in the protein backbone are taken into account in b-phipsi and w-rdist for efficiency and robustness.

Surface triangles difference of molecular alpha shapes (t-alpha) is a metric that relies on the number of triangles found by Delaunay Triangulation that cover an alpha shape[77], which engulfs a point cloud from the normalized atomic coordinates of a protein. The value of the alpha parameter for the computation of t-alpha is set to 0.085 based on manual tuning on a set of homologous protein structures in different resolutions. The number of triangles ($T_A$, $T_B$) that cover the surfaces of the generated alpha shapes is the core part of this metric:

$$t_{alpha} = e^{\left|\log(T_A)-\log(T_B)\right|} - 1$$

(3)

**Constrained mode search on domains.** The protein domain comparisons rely on available domain information from UniProt[78] and RCSB PDB data. These data include the range of the residue positions of each present domain in a candidate protein structure. Each of the domains is compared with the ones of the reference protein in separate paired sessions. The results contain information on the compared domains.

**Constrained mode search on binding sites.** This mode requires the segments of interest as input. The positions of the reference protein's interacting residues are converted into multiple contiguous residue ranges by Gaussian Mixture Models (GMM) clustering supported by Silhouette Analysis[79]. The spatial outliers are discarded based on the interquartile range (IQR). Hydropathy Index[80] is grouped into distinct classes of neighboring values (Supplementary Table 5) whose class labels substitute the letter codes in the protein sequence that is retrieved from PDB data. A mixed representation is constructed by mapping a secondary structure state (estimated with the DSSP[81] method from PDB data or with STRIDE[82] as a fallback for non-standard formatted PDBs) and a hydrophobicity label to an ASCII character for both reference and candidate proteins. Separate local alignments between the contiguous reference segment parts and the candidate's sequence define the areas on which the metrics are going to be computed (match score = 5, mismatch score = −4, open gap penalty = −10, gap extension penalty = −0,5). Overlapping aligned ranges are unified through an iterative process, eliminating any falsely over-represented intersecting areas. The final results also include full information on the alignment of each candidate protein for further inspection. Alternatively, there is an option for the user to choose 1D, 2D or Hydrophobicity sequence alignments instead.

Below there is an example of the mixed sequence alignment between a part of a Spike protein's predicted binding site (positions 744–774) and ACE2 (positions 545–572) based on PDB data (6VXX.A[29], 3D0H.B[83]):

- Protein primary structure sequences:
  GDSTEJSNLLLQYGSFITQLNRALTGIAVEQ,
  SNSTEAGQKLFNMLRLGKSEPWTLALEN.
- Sequence of hydrophobicity group labels:
  34334634111433320341452133020 44,
  34334234412421513434333312144.
- Protein secondary structure sequences:
  TT.HHHHHHGGGGGGTHHHHHHHHHHHHH,
  TT.HHHHHHHHHHHTTTTSS.HHHHHH.
- The alignment of mixed representations (ASCII), mapping hydropathy cluster labels to secondary structure states:

```
NO}$%'$%""7:9998 K$%"%&#"-------$$!#!%%
|||||.|| ||..|| ||.|.||
NO}$%#$%----------%"#%#"PLNOUV}$$"#"%%
```

**Selecting the structurally similar candidates.** The candidates are filtered, discarding samples by the PDB metadata section ('organism scientific', 'gene') and user-specified constraints. Depending on the dataset's total size, 1% of the top rows is sampled for each metric and combined into a new set with duplicates removed. The samples are ordered by the Borda Count rank aggregation algorithm and are clustered by HDBSCAN clustering algorithm[84] (min_samples = 5 and Euclidean distance metric) (Supplementary Fig. 16). The ordered sample set is traversed until a non-noisy sample is encountered, whose label determines the preferred cluster, preserving the maximum set between the top 15% of the previously ranked samples or the previously traversed samples. On a failed attempt, the minimum cluster size is lowered until a pre-defined number of attempts is exceeded, and the clustering result is discarded if the cluster size is below a specified threshold. Samples clustered with a probability below 0.1 are discarded as noise, and the results are re-ranked and truncated either to a maximum of 800 top-ordered entries or to 250 entries if the dataset is non-redundant (option is configurable). All intermediate results are stored. Visualizations are generated using the Uniform Manifold Approximation and Projection (UMAP) method, displaying the clustered space of the results.

**Enrichment and assessment of the selected candidate entries.** The resulting list of the selected proteins is enriched with information from UniProt's ID mapping offline resource and from RCSB PDB GraphQL, UniProt, NCBI Entrez[85] and EBI QuickGo[86] online web services. Duplicate protein or gene entries are discarded, keeping the entry with the highest order by the metrics. An evaluation module performs global sequence alignments, determines the sequence identity, computes the 3D similarity with a modified version of TM-Align and calculates chemical similarity for each of the proteins in the final sample. The alignments (match score = 5/mismatch score = −4 similar to EDNAMAT/NCBI NUC 4.2 scoring matrix, open gap penalty = −10, gap extension penalty = −0,5, no end-gap penalty) take place between 1D/2D partial data derived from PDBs (leaving a gap between non-consecutive residues), full 1D sequences from UniProt (matches/mismatching scoring by BLOSUM62 matrix[87]) and RNA sequences from NCBI RefSeq[88]. The sequence identity is calculated both with gaps excluded and included and unless stated otherwise, the sequence identity metric displayed in the rest of the

article is the one with the gaps included:

$$identity = \frac{matches}{matches + mismatches + gaps} \quad (4)$$

and for gaps excluded identity:

$$identity_{gaps} = \frac{matches}{matches + mismatches} \quad (5)$$

Finally, these computations are appended to the results, and the combined output, along with aggregated information on Gene Ontology terms, is exported to a UniProt data-enriched, printer-friendly HTML report by a separate presentation module. This module also generates lineage trees, word cloud (Supplementary Fig. 17) visualizations, protein secondary structure alignments/content plots and statistical radial plots based on the proteins in the results of a search session.

**Structure-localized suggestions for novel properties.** First, the 2D alignments match per position of any proteins in the results, associated with user-chosen GO properties or search terms, are summed. Alignments of 1D, mixed or hydrophobicity sequences can be chosen instead. The resulting curve is treated as an information signal for whom an exponential weighted moving average (EWMA) is calculated with a span equal to 30% of the reference protein sequence length. The peaks ($P_S$) are selected by the following operation:

$$P_S = P_T \cup P_D, D = \mu_\epsilon + \sigma_\epsilon \quad (6)$$

Where $P_T$ are the peaks with top 10% matches, $P_D$ are the peaks with a distance greater than $D$, $\mu_\epsilon$, $\sigma_\epsilon$ the mean and standard deviation of the distances between all the peaks and EWMA. This computed peak prominence leads to the correlation of the assessed desired property with a set of residue positions in the reference protein structure. Log files are constructed that contain information on the alignments.

**Prediction of Spike protein's binding sites.** The SARS-CoV-2 S protein structure (PDB ID: 6VXX) was manually processed via the Schrodinger Maestro Suite with the Protein Preparation Wizard[89] removing ligands and water molecules, filling missing residues with Prime[90], determining protonation states in physiological pH (7.4) with PropKa[91] and minimized with OPLS force field[92] (Ramachandran and Janin plots: Supplementary Figs. 18 and 19). The binding sites were predicted using the SiteMap module[93] (Supplementary Table 6).

**Reporting summary.** Further information on research design is available in the Nature Portfolio Reporting Summary linked to this article.

## Data availability

Source data from all figures are available in Supplementary Data 1. The utilized UniProt ID mapping resources were retrieved on 21/9/2021 at: https://ftp.uniprot.org/pub/databases/uniprot/current_release/knowledgebase/idmapping and RefSeq resources were retrieved on 6/11/2021 at: ftp://ftp.ncbi.nlm.nih.gov/refseq/. RCSB PDB GraphQL (https://data.rcsb.org/graphql), UniProt (https://www.uniprot.org/uploadlists/, https://www.uniprot.org/uniprot/ACCESSION.xml/<ACCESSION-NUMBER>) and EBI QuickGo (https://www.ebi.ac.uk/QuickGO/services/ontology/go/terms/GO:<termid>) online services are used as a fallback method to retrieve required data which are not present in the local static data sources. The viral PDB dataset for the comparisons to Spike was obtained in December 2020 and it was the query result for viral proteins in RCSB (https://www.rcsb.org/docs/programmatic-access/batch-downloads-with-shell-script). The human PDB dataset was assembled in a similar way on 19/6/2022 by querying RCSB PDB for all the available PDBs that contain human proteins. The dataset containing PDB files of the Spike protein was obtained from the same source (RCSB). The predicted human protein dataset corresponds to the predicted human protein by AlphaFold v4 (https://ftp.ebi.ac.uk/pub/databases/alphafold/latest/UP000005640_9606_HUMAN_v4.tar). Benchmark datasets were obtained from https://github.com/rcsb/biozernike-validation for Task 1, http://shrec2018.drugdesign.fr/shape_retrieval_shrec2018_pdb_files.tar.gz, http://shrec2018.drugdesign.fr/SHREC2018_ref.cla for Task 2 and http://ekhidna2.biocenter.helsinki.fi/dali/pdb_and_scope.tar for Task 3.

## Code availability

Machaon's implementation and evaluation scripts are available to the community at: https://github.com/anastasiadoulab/machaon. MachaonWeb's implementation is also available at: https://github.com/anastasiadoulab/machaonweb. The computed features on the PDB files in Spike protein structural comparisons with viral dataset are available at https://zenodo.org/record/6654658. Machaon was implemented in Python v.3.8 (https://www.python.org/) and uses the following packages: NumPy (https://github.com/numpy/numpy), pandas (https://github.com/pandas-dev/pandas), SciPy (https://www.scipy.org/), scikit-learn (https://github.com/scikit-learn/scikit-learn), Matplotlib (https://github.com/matplotlib/matplotlib), seaborn (https://github.com/mwaskom/seaborn), BioPython (https://github.com/biopython/biopython), RDKit (https://github.com/rdkit/rdkit), Open3D (https://github.com/isl-org/Open3D), HDBScan (https://github.com/scikit-learn-contrib/hdbscan), ranky (https://github.com/Didayolo/ranky), DSSPParser (https://github.com/neolei/DSSPparser), Pebble (https://github.com/noxdafox/pebble), protobuf (https://github.com/protocolbuffers/protobuf), lxml (https://github.com/lxml/lxml), networkx (https://github.com/networkx/networkx), Plotly (https://github.com/plotly/plotly.py), UMAP (https://github.com/lmcinnes/umap), word_cloud (https://github.com/amueller/word_cloud), tqdm (https://github.com/tqdm/tqdm), beautiful soup (https://code.launchpad.net/beautifulsoup). A modified version of TM-align (https://zhanglab.dcmb.med.umich.edu/TM-align/, source code retrieved on 10/2/2021) is used by the evaluation model for 3D structure similarity computation. Secondary structures are determined by DSSP (https://github.com/PDB-REDO/dssp). The distributed computing platform MachaonWeb was implemented in Rust v.1.68 (https://www.rust-lang.org), React Framework v.18.2.0 (https://github.com/facebook/react) and its storage depends on MariaDB v.10.11.2 (https://mariadb.org). It is deployed via Docker (https://www.docker.com) and uses the following packages: Rust-based: async-stream (https://github.com/tokio-rs/async-stream), anyhow (https://github.com/dtolnay/anyhow), axum(https://github.com/tokio-rs/axum), axum-server (https://github.com/programatik29/axum-server), chrono (https://github.com/chronotope/chrono), diesel (https://github.com/diesel-rs/diesel), dotenvy (https://github.com/allan2/dotenvy), futures (https://github.com/rust-lang/futures-rs), glob(https://github.com/rust-lang/glob), prost (https://github.com/tokio-rs/prost), rand(https://github.com/rust-random/rand), regex(https://github.com/rust-lang/regex), reqwest (https://github.com/seanmonstar/reqwest), rustls (https://github.com/rustls/rustls), serde (https://github.com/serde-rs/serde), serde_json(https://github.com/serde-rs/json), sha2 (https://github.com/RustCrypto), tokio (https://github.com/tokio-rs/tokio), tonic (https://github.com/hyperium/tonic), tower (https://github.com/tower-rs/tower), tracing (https://github.com/tokio-rs/tracing), unicode-segmentation (https://github.com/unicode-rs/unicode-segmentation), uuid (https://github.com/uuid-rs/uuid), walkdir (https://github.com/BurntSushi/walkdir). Javascript-based: axios (https://github.com/axios/axios), bootstrap (https://github.com/twbs/bootstrap), react-bootstrap (https://github.com/react-bootstrap/react-bootstrap), react-copy-to-clipboard (https://github.com/nkbt/react-copy-to-clipboard), react-ga4 (https://github.com/codler/react-ga4), react-google-recaptcha-v3 (https://github.com/t49tran/react-google-recaptcha-v3), react-router-bootstrap (https://github.com/react-bootstrap/react-router-bootstrap), react-router-dom (https://github.com/remix-run/react-router), react-scripts (https://github.com/facebook/create-react-app), web-vitals (https://github.com/GoogleChrome/web-vitals). Machaon communicates with MachaonWeb via an mTLS-based gRPC module implemented in Python v.3.8 and uses the Python implementation of gRPC library (https://github.com/grpc/grpc).

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

## Acknowledgements

We thank Schrödinger Inc. for providing access to the Maestro suite via academic licensing. We thank Dr. Ioannis Michalopoulos (Biomedical Research Foundation, Academy of Athens, Greece) for his advice and communications on the online hosting of MachaonWeb in HYPATIA cloud infrastructure. HYPATIA has been funded by the 'ELIXIR-GR: Managing and Analysing Life Sciences Data (MIS: 5002780)' project (co-funded by Greece and the European Union - European Regional Development Fund). Publication fees are covered by the National Public Investment Program of the Ministry of Development and Investment/General Secretariat for Research and Technology (Greece), in the framework of the Flagship Initiative to address SARS-CoV-2.

## Author contributions

Conceptualization, design and implementation of the method: P.K. Organizing and writing the manuscript: P.K. and E.A. Computational consulting: I.E. and I.V. Biological consulting: D.T. and E.A. Final editing: P.K., E.A., I.E., I.V., D.T., G.L.B.

## Competing interests

The authors declare no competing interests.
