## [Peer Review File · Communications Biology]

Reviewers' comments:

Reviewer #1 (Remarks to the Author):

In "Identifying and profiling structural similarities between Spike of SARS CoV-2 and other viral or host proteins with Machon" Kakoulidis et al., suggest a new in silico tool to search structurally related proteins for a protein of interest. They use torsion angles, amino acid distances and protein shape as measures to compare protein characteristics in a multivariate statistical analysis (here 3D). While I find the approach appealing, I have some serious doubts that require detailed answering. When inserting a reference structure, this only represents a snapshot of the protein in a very restricted conformational orientation. For the SARS-CoV2 S-protein, multiple conformational states were reported (e.g. open and close conformation, bound and unbound). Making use of measures that define protein flexibility (torsion angles, distances and general shape) is in my opinion very likely to produce artefacts and should be prone to bias. The SARS-CoV2 S-protein has a very different shape in open and close conformation and distances/angles of residues interacting with ACE2 change dramatically upon binding. However, the core protein structure remains stable and I would guess that Machon is biased by these core protein structures and will miss many interesting (flexible) protein features. What are measures during selection of different SARS-CoV2 S-proteins as user input? You selected the A-chain of a wt (PDB: 6VXX), Delta (PDB:7V7Q) and Omicron (PDB:7T9K), however, all three structures were derived from different symmetry (wt = C3; Delta = C1) or occupation states (Omicron bound to ACE2). While you show all results that come up with Machon, you fail to show the robustness of your algorithm. How biased are results by the input conformation? When you discuss evolutionary structural conservation you must show that this is independent of the conformation. Also, only because structures share common features, does not mean anything. Show specific examples that demonstrate functional equivalents across large evolutionary distances that have been preserved and can be picked up by Machon. Overall, you correlate large datasets with large datasets and to identify similarities is not surprising! Why would it be meaningful to compare S-proteins with the "ubiquitin" GO term? What does it tell me? Together, you fail to demonstrate robustness against user input and you fail to demonstrate specific and precise examples that reveal biological meaningful similarities between proteins of distant relation.

Major points:

1. Use different conformational orientations of the same protein (e.g SARS-CoV2 S-protein, ABC transporters, Hemoglobin) as input and show that Machon identifies the same proteins from a dataset as similar. Report how many hits were identical (also in hierarchy) between the different conformations of the same input protein.
2. Identify a specific biological example where Machon identifies similar protein functions in very distantly related proteins that has been unknown/overlooked so far. E.g. does Machon identify those helical parts of S-proteins from different viruses as similar that after release insert into the host membrane)?
3. Report on the secondary structure element distribution of similarities identified by Machon. How many percent are α -helical, β -sheet or loop region? Where are most similarities found? Protein core or surface?
4. Contributions statement: four supervisors for one PhD or PostDoc are too many! Please specify the exact role of each supervisor or remove him or her from the manuscript!

Reviewer #3 (Remarks to the Author):

Kakoulidis et al., present a method called Machaon that utilizes protein structure based calculations such as phi/psi, interresidue contacts, and surface complexity. The authors have applied this method on spike protein, and its two variants. While this is the first impression of the paper, it was clearly not as presented in the Results section. After many readings it was clear that the focus is not on spike protein but rather the tool itself. In my opinion this is an excellent work for computational journals where presentation of the method and its key features are described in detail. The tool is available in github which is rarely used by biologists and requires more knowledge of Linux. While DALI server is easily accessible on website. Thus, the manuscript at present stands unacceptable for broader audience. Few points that may help authors to submit it again in another targeted journal.

For instance the earlier existing DALi server targets spatial restraints, however presented method utilizes a novel approach on micro and macro properties of structure. I am also not sure if combining side-chain and surface area is a productive idea. Did authors obtain key regions due to high angle similarity?

To show application, authors have primarily utilized spike protein but in opinion the data is quite large and would bias heavily their results. Proper analysis on globular proteins, and other differently shaped proteins will help reader understand more applications of Macheon.

Lastly the tool may be available as website for utilization by biologists.

Reviewers' comments

Reviewer #1 (Remarks to the Author):

In “Identifying and profiling structural similarities between Spike of SARS CoV-2 and other viral or host proteins with Machon” Kakoulidis et al., suggest a new in silico tool to search structurally related proteins for a protein of interest. They use torsion angles, amino acid distances and protein shape as measures to compare protein characteristics in a multivariate statistical analysis (here 3D). While I find the approach appealing, I have some serious doubts that require detailed answering.

When inserting a reference structure, this only represents a snapshot of the protein in a very restricted conformational orientation. For the SARS-CoV2 S-protein, multiple conformational states were reported (e.g. open and close conformation, bound and unbound). Making use of measures that define protein flexibility (torsion angles, distances and general shape) is in my opinion very likely to produce artefacts and should be prone to bias. The SARS-CoV2 S-protein has a very different shape in open and close conformation and distances/angles of residues interacting with ACE2 change dramatically upon binding. However, the core protein structure remains stable and I would guess that Machon is biased by these core protein structures and will miss many interesting (flexible) protein features. What are measures during selection of different SARS-CoV2 S-proteins as user input? You selected the A-chain of a wt (PDB: 6VXX), Delta (PDB:7V7Q) and Omicron (PDB:7T9K), however, all three structures were derived from different symmetry (wt = C3; Delta = C1) or occupation states (Omicron bound to ACE2). While you show all results that come up with Machon, you fail to show the robustness of your algorithm. **How biased are results by the input conformation?** When you discuss evolutionary structural conservation you must show that **this is independent of the conformation**. Also, only because structures share common features, **does not mean anything**. Show **specific examples** that demonstrate functional equivalents across large evolutionary distances that have been preserved and can be picked up by Machon. Overall, you correlate large datasets with large datasets and to identify similarities is not surprising! **Why would it be meaningful to compare S-proteins with the “ubiquitin” GO term? What does it tell me?**

Major points:

1. Use different conformational orientations of the same protein (e.g. SARS-CoV2 S-protein, ABC transporters, Hemoglobin) as input and show that Machon identifies the same proteins from a dataset as similar. Report how many hits were identical (also in hierarchy) between the different conformations of the same input protein.

Response:

In the manuscript, we set one of the monomers in the closed state of the Spike protein structure as a reference since we aimed to assess the protein prior to its attachment to the cell. Regarding the monomers from Delta and Omicron variants, they adopt a very similar conformation, as it is shown in figure 3A. Thus, our results are not impacted by the differences of the quaternary structure. For clarification, we now mention in more places of the manuscript that we use Spike protein monomer for the comparisons. Also, in Results - “Seeking biases in the results for SARS-CoV-2 Spike monomer”, we refer to the method’s metrics for different versions of SARS-CoV-2 Spike protein’s monomer structure (“*alternative resolutions, conformations, lengths and variations*”). Our analysis showed that these versions would rank high in the results if they were of unknown origin due to their low median metric values. B-

phipsi and w-rdist had lower median values than the median ones of the Spike protein’s results on the large dataset of ~40000 structures. Median t-alpha values were also close: “The median values of the metric vector [B-phipsi, W-rdist, T-alpha] in the final cluster were [0.0180, 0.9410, 0.0378] and for the Spike proteins set were [0.0025, 0.7334, 0.0666] (Supplementary Data 4).”

Following the suggestions in the review, we performed Normal Mode Analysis (NMA) with R and Bio3D package to generate different conformations of Spike protein’s monomer in open and closed states (34 conformations for each), selecting a protruding monomer for the case of open state. We used preprocessed structures for NMA, following the data preparation procedure that is described in Methods - “Prediction of Spike protein’s binding sites“. We aggregated the method’s outputs targeting the viral dataset of ~40000 structures, for each of the conformations (68 total). We measured the percentage of the common proteins in the results by their UniProt ID (the ones without a mapped ID were not taken into account). PDB IDs were also counted to determine the exact match of the structures in the results. According to the reviewer’s comment, we repeated these computations, using the same procedure, for additional proteins: an ABC transporter, ATP-dependent translocase ABCB1 (AlphaFold filename: AF-P08183-F1-model_v4), Hemoglobin A (oxy [PDB ID: 2DN1.A], deoxy [PDB ID: 2DN2.A], carbonmonoxy forms [PDB ID: 2DN3.A]), Hemoglobin B (oxy [PDB ID: 2DN1. B], deoxy [PDB ID: 2DN2. B], carbonmonoxy [PDB ID: 2DN3.B] forms) and Hemoglobin A & B (oxy [PDB ID: 2DN1], deoxy forms [PDB ID: 2DN2] / merged PDB chains of A & B). The PDB files (except for the AlphaFold structure) were retrieved from PDB Redo Databank (<https://pdb-redo.eu/>) and all the reference structures were preprocessed as in the case of open/closed state Spike monomers. Machaon’s results for the top 100 unique protein entries are shown below:

Protein	Common by UniProt		Absolute position difference		Common by PDB ID		Absolute position difference	
	(Median %)	(STD %)	Median	STD	(Median %)	(STD %)	Median	STD
Spike	66	11	12	5	34	17	15	7
ABCB1	57	14	6	4	49	23	6	5
Hemoglobin A	40	14	16	6	31	16	16	6
Hemoglobin B	46	15	16	7	32	19	15	7
Hemoglobin A&B merged	32	17	22	11	15	18	22	13

We observe that SARS-CoV-2 Spike has the most consistency in its unique protein entries (largest median, lowest standard deviation), since the search space contains a high number of viral proteins. The rest of the proteins maintain unique entries across their conformational changes according to their chain length. For instance, ABC transporter is a large protein with 1280 aminoacids and larger stable areas than Hemoglobin A & B, each one including less than 150 aminoacids. In the case of the merged Hemoglobins, there is less consistency in the identified structurally similar proteins between each conformation due to the higher flexibility of the complex. Regarding the measurements on PDB IDs, we can see that the

percentages are lower with greater variability (larger STD) because on each conformation a different PDB file is picked for the same protein. The target dataset is redundant and as a result some proteins are represented by more than one PDB file. As we mention in the manuscript (Discussion, second paragraph), PDB files might be missing some parts, vary in quality or contain a different conformation, thus we did not design a non-redundant dataset. The method utilizes all the available data and any overrepresentations are handled by HDBSCAN algorithm which combines hierarchical and density-based clustering.

All relevant data and scripts are available for review in:
https://machaonweb.com/reviews/nma_results.zip (4.6 GB file)

From these results, we can see that the method adapts to each conformation since a folding change affects the angles, the inter-residue distances and the overall surface of a protein. A certain percentage of the proteins in the results remains the same as the detection of the structural similarities adheres to parts of the reference protein's structure that are less affected by the conformational change. These could be areas of the protein's core or functional sites whose local structures must be maintained in order to preserve the associated biological processes of the organism. Regarding the ranking differences of the same proteins among the results, it is something expected as the folding changes. As stated in Discussion: *"The novelty of Machaon approach is that it regards each protein as an assembly of flexible moving parts and not a rigid hollow solid structure. It considers the arrangement of the protein's inner components (B-phi-psi, W-rdist) and how this is reflected to the resulting surface (T-alpha)".* We offer a different and complementary perspective on the problem of structural similarity by not focusing only on similarities that are invariant to conformational change or high structural similarity. Machaon is shown that operates beyond domain and folding classes (Results, "Validation of the method's accuracy", page 3).

In the manuscript, we endorse a cited study's observation on the difficulty of the structural comparison's evaluation: *"The interpretation of our tests emphasizes on the structural similarity of the identified proteins which is an effective validation strategy since there is not a universal ground truth available²⁷".* It would have been really fortunate if there was a clear reference or an established cut-off of what is noise or not, due to the existence of functional sites of few residues such as the case of DEDH catalytic tetrad of AGO2 protein or short linear motifs in disordered regions (SLiMs, mentioned in the end of "Investigating the biological properties of SARS-CoV-2 Spike protein" section). Structural similarity does not only support a hypothesis of an evolutionary relationship, but also provides key information for purposes like drug targeting or drug repurposing, which are suggested as additional potential areas of application in the introduction of the manuscript. The knowledge of structural similarity could be incorporated in the process of biomarker discovery or in the determination of concealed virus-host co-evolutionary relationships. In our most recent results targeting the AlphaFold DB dataset, a top-scoring entry was thyroglobulin, which is presented as a biomarker in literature. Also, the structural comparison depends on the reference protein and the proteins that will be compared. If the search space consists of proteins with very different structures or the reference protein is chosen to be a challenging one with disordered parts, the method will adapt and will attempt to identify whatever similarity is available.

2. Identify a specific biological example where Machon identifies similar protein functions in very distantly related proteins that has been unknown/overlooked so far. E.g. does Machon identify those helical parts of S-proteins from different viruses as similar that after release insert into the host membrane)?

Response:

We provide such examples in the manuscript where we connected the proteins in the results with Gene Ontology terms in order to offer indications of potential novel relationships with molecular functions, cellular components and biological processes. For instance, we performed meta-analysis on the proteins in the results for ubiquitination, which is an essential process for viral life cycle and replication. Viruses have been observed to participate in this pathway in order to regulate the mechanisms of the host and evade its immune system. For convenience and clarification, we added a table in the Supplementary Information (Supplementary Table 1) that presents the proteins that were found to be associated with Gene Ontology terms that contain “**ubiquit**” substring.

A few months after the manuscript’s submission, a study was published in Nature’s Signal Transduction and Targeted Therapy journal that confirmed the connection between SARS-CoV-2 Spike and the biological process of ubiquitination [1]. More specifically, the authors found that the virus hijacks ubiquitination mechanisms for its own advantage and they discovered three ubiquitination sites in Spike protein. This development is in accordance with the method’s indications regarding the relationship of Spike and ubiquitination.

3. Report on the secondary structure element distribution of similarities identified by Machon. How many percent are α -helical, β -sheet or loop region? Where are most similarities found? Protein core or surface?

Response:

To address the above comment, a new visualization was added to the set of Machaon’s data visualizations along with alignment content logging. We measured the 2D aligned content of the finalist proteins that are identified by the method on the whole structure search for structural similarity with Spike protein monomer. Spike protein monomer. We observed that the method does not focus on a specific secondary structure unit (e.g. α -helix), but adapts according to the reference protein and the search space. For example, in Spike’s case we can see that the percentage of the aligned α -helices and beta-sheets is almost equal in the results for viral dataset (Supplementary Figures 13). However, in the results regarding the human datasets the patterns are diverse with different sets of aligned 2D folding types (Supplementary Figures 14-15). Based on these measurements (“Seeking biases in the results for SARS-CoV-2 Spike monomer”, end of last paragraph, page 8) and the results presented in the manuscript (“Validation of method’s accuracy”, page 3), Machaon is not restricted to detect similarity on a particular folding type, folding class or domain class. This is further confirmed in the results of other human and viral proteins that are listed in the Sample Results section of MachaonWeb, the online platform that was created for the web access of this method (<https://machaonweb.com/results>).

The core metrics of the method are not based on one-to-one alignments, so there is no hard constraint that dictates the identification of high and contiguous structural similarity. The identified structural similar content might be non-contiguous and dispersed as mentioned in the manuscript (“Validation of method’s accuracy” section at the beginning, page 3). B-phiPsi and w-rDist metrics are related to the statistically modeling of the actual values of phi-psi angle pairs and inter-residue distances by distributions. Adding T-alpha metric as a geometrical filter, we define a space where molecules could fit in without extruding. Therefore, the combination of these metrics describes how well a molecule could fit in a space of a reference shape and how close is its composition (phi-psi angles, inter-residue distances) to a reference composition. It is a non-linear comparison that considers the multi-dimensional aspects of a molecule and could be applied beyond proteins such as RNAs or even non-organic molecules. For this reason, it is not

directly comparable with other methods as mentioned in the manuscript (“Validation of method’s accuracy” section, first paragraph, page 3).

We also extended further the set of Machaon’s data visualizations with a plot on the protein secondary alignments between all proteins in the results and the reference protein (mentioned in Methods, “Enrichment and assessment of the selected candidate entries”). The plot for Spike protein illustrates the spreading of the structural similarities throughout the extend of the peptidic chain (Figure 9). This confirms that the method does not emphasize on a particular part of the protein structure and it attempts to match whatever is available in the target dataset. There are plots available for other proteins in the Sample Results section of MachaonWeb (<https://machaonweb.com/results>).

4. Contributions statement: four supervisors for one PhD or PostDoc are too many! Please specify the exact role of each supervisor or remove him or her from the manuscript!

Response:

The authors’ contribution is elaborated in the appropriate section and the role of each supervisor was specified (see “Author Contributions” section in the revised manuscript).

Reviewer #3 (Remarks to the Author):

Kakoulidis et al., present a method called Machaon that utilizes protein structure based calculations such as phi/psi, interresidue contacts, and surface complexity.

1.The authors have applied this method on spike protein, and its two variants. While this is the first impression of the paper, it was clearly not as presented in the Results section. **After many readings it was clear that the focus is not on spike protein but rather the tool itself.**

Response:

Reviewer #3 correctly pointed out that Machaon’s development possesses a central role in our work. However, the manuscript extends beyond the plain description of the methodology, as it was applied to study the Spike protein. Specifically, we reported the results of Machaon’s application using three versions of the Spike protein monomer (native, Delta, Omicron) and a large dataset containing PDBs with viral proteins. We performed comparisons in whole structures, domains and segments of choice, looking for proteins with structural similarities in each of the levels. Specifically for segment scanning, we initially predicted potential binding sites (Supplementary Table 6) after preprocessing the native Spike monomer structure, following a Molecular Dynamics data preparation protocol (“Prediction of Spike protein’s binding sites”, Methods) and the resulting structure is provided in the Source Data. We use the predicted binding sites as reference to scan for proteins that contain similar segments. Additionally, we perform pairwise secondary structure alignments between the three Spike monomer variants (Supplementary Figure 11) to showcase their 2D structural differences. Also, we created custom versions of the corresponding PDB chains of the three monomers in order to have the same missing residue positions for an accurate comparison of the variants. The final sets of each search session contain proteins with similarities on protein sequence, genomic, chemical and Gene Ontology levels as revealed by Machaon’s

meta-analysis. All results were connected with the latest available literature during the writing of the manuscript and they were available as Supplementary Data

We also extended our comparisons for Spike protein on two large datasets that mostly consist of human protein structures: one with experimental data from RCSB PDB (~160000 PDB chains) and one with predicted data from AlphaFold DB (~23000 PDB chains). The results are presented in the last paragraph of “Identifying structurally similar proteins to SARS-CoV-2 Spike monomer”, page 6 and discussed in Discussion, page 10 at the center. This work provides more evidence on the viral mimicry of SARS-CoV-2 and most of the identified proteins are correlated with the virus by literature. Some of these correlations have been published after the submission of this manuscript, validating further the proposed technique such as the cited papers #45, #46, #47, #48 and #49. Also, we observed that the newer results complement previous findings on the relationship of Spike protein with ubiquitination (“Investigating the biological properties of SARS-CoV-2 Spike monomer”, first paragraph, page 6). Supplementary Data, in the revised manuscript, include the whole results of these searches for further review and research (Supplementary Data 4 & 5). In any case, the specific journal, Communications Biology, had previously accepted and published computational manuscripts [2].

2. In my opinion this is an excellent work for computational journals where presentation of the method and its key features are described in detail. The tool is available in github which is rarely used by biologists and requires more knowledge of Linux. While DALI server is easily accessible on website. Thus, the manuscript at present stands unacceptable for broader audience. Few points that may help authors to submit it again in another targeted journal.

An alternative and more user-friendly access was developed and currently available through our distributed computing platform MachaonWeb, <https://machaonweb.com>. For more details please check Reviewer #3, point 5.

3. Few points that may help authors to submit it again in another targeted journal. For instance the earlier existing DALI server targets spatial restraints, however presented method utilizes a novel approach on micro and macro properties of structure. I am also not sure if combining side-chain and surface area is a productive idea. **Did authors obtain key regions due to high angle similarity?**

Response:

As stated in the response for point 1 of Reviewer #1, Machaon cannot be directly comparable to existing methods such as DALI. Please check the first paragraph of “Validation of the method’s accuracy” in the page 3 of the manuscript for more details. The intention of the Machaon is to become an open suite that the results of other methods could be incorporated in it, especially in the stage of meta-analysis, where several established metrics have already been integrated such as TM-Score and Tanimoto Index.

We believe that we generated a robust method, as indicated by the tasks performed, by incorporating the metrics described below:

- **B-phipsi:** This is the distance between the two distributions of phi-psi angle pairs, one distribution per protein. These distributions model how probable is an angle pair of certain values to occur in a protein. Therefore, this metric quantifies how close is for two compared proteins to have the same occurrence of phi-psi angle pair values in their peptidic chains.

- **W-rdist:** This is the distance between the two distributions of inter-residue distances, one distribution per protein. These distributions model how probable are inter-residue distances of certain values to occur in a protein. Therefore, this metric quantifies how close is for two compared proteins to have the same occurrence of inter-residue distance values in their peptidic chains.
- **T-alpha:** This is a difference on the triangulation of two compared protein surfaces and it is based on the geometric theory of alpha shapes. This metric measures how far are the numbers of the triangles that cover the surface of each protein. The computations take place after bringing all the atomic coordinates of each protein to the same scale via normalization. Therefore, it quantifies the surface complexity difference through triangulation without taking into account the difference in size.

We provide additional description of the metrics in the Discussion of the revised manuscript, first paragraph, page 8. All metrics do not perform one-to-one comparisons and they do not rely on any kind of alignments. Alignments are only employed as a pruning step on segment scanning, where they take place between mixed representations of structural and hydrophobicity information. Through these approaches, we manage to generate information regarding key regions in domain-level or segment-level (“Identifying structurally similar proteins to SARS-CoV-2 Spike monomer”, third paragraph, page 5).

4. To show application, authors have primarily utilized spike protein but **in opinion the data is quite large and would bias heavily their results.** Proper analysis on globular proteins, and other differently shaped proteins will help reader understand more applications of Macheon.

Response:

The application of Machaon to the monomer of the SARS-CoV-2 Spike protein was carried out to mainly detect structurally similar viral proteins. It was a process similar to Virtual Screening where a large database of ligand structures is assessed for their best fit in a number of criteria. We have several mechanisms in place to combat statistical outliers in large datasets with sampling and noise filtering (as described in Methods, “Selecting the structurally similar candidates”).

In the revised manuscript we test how the method performs in three tests on different public datasets with various sizes:

- BioZernike’s validation set: 2685 structures belonging to 151 CATH families. This test verified that the method operates beyond the CATH classification.
- SHREC 2018 challenge dataset: 2267 structures deriving from the conformational space of 107 proteins. The method identified the majority of alternative conformations of the same protein but also other proteins with comparable structural similarity.
- SCOP140 benchmark 15211 structures that represents the classified proteins in SCOPe 2.07. This test verified that the method operates beyond the SCOP classification.

Based on the review’s comment, we generated results for more proteins and on different target spaces proteins as presented in the Sample Results section of MachaonWeb (<https://machaonweb.com/results>), the online platform of the method. We applied Machaon on human proteins that are encoded by genes of high research interest and viral proteins from known viruses:

[Human]

- Epidermal growth factor receptor (EGFR)
- Methylenetetrahydrofolate Reductase (MTHFR)
- Tumor Necrosis Factor (TNF)
- Tumor Protein P53 (TP53)
- Vascular Endothelial Growth Factor A (VEGFA)

[Viral]

- Ebolavirus nucleoprotein
- Epstein-Barr virus (EBV) Major capsid protein
- Human immunodeficiency virus 1 (HIV-1) capsid protein (p24)
- Human papillomavirus type 16 (HPV16) Major capsid protein L1
- Monkeypox Virus (MPXV) E4
- Severe acute respiratory syndrome coronavirus 2 (SARS-CoV-2) Spike protein

Each one of these proteins were compared with three large datasets (separate sessions) on whole structure comparisons:

- The viral dataset that is used in the study of Spike protein
- A redundant dataset that mainly consists of human proteins
- The predicted human proteome by AlphaFold

The method was also executed for multiple conformations of an ABC transporter and hemoglobin as was suggested by Reviewer #1, point 1. The results of each conformation are also available for review: https://machaonweb.com/reviews/nma_results.zip (4.6 GB file).

We expect that this extensive additional, to the original manuscript, analysis demonstrates the application of Machaon beyond the spike protein monomer as protein structures of different origin, size shape and function were also used.

5. Lastly the tool may be available as website for utilization by biologists

Response:

The method is implemented in a cross-platform programming language (Python) and it can be executed in any platform except for its meta-analysis part that relies on Unix-based functions. Additionally, we offer the solution of Docker or Singularity for easy deployment in any platform or High Performance Computing (HPC) infrastructure. Taking into account the reviewer's comments, we prepared an alternative and more user-friendly access via our distributed computing platform MachaonWeb, <https://machaonweb.com>. The platform is written in Rust programming language and supports the majority of the method's capabilities in a simplified interface for increased productivity. MachaonWeb includes a section on brief and detailed instructions for an easy onboarding of the new users. The user is able to choose a reference protein structure and define the search space of the method in any of the three comparison modes available (whole, domain, segment). We increase the options of the user as we support predicted

structures from AlphaFold and ESM Metagenomic Atlas. The platform has a section where multiple sample results are provided on biologically significant proteins (whole structure comparisons). Finally, Machaon (the core method) was further optimized and simplified for higher efficiency in usage and deployment. The codes of the latest version of Machaon and MachaonWeb will be available to the community as open source software, allowing on-site installation wherever needed.

[1] Xu, G. et al. (2022) "Multiomics approach reveals the ubiquitination-specific processes hijacked by SARS-COV-2," *Signal Transduction and Targeted Therapy*, 7(1).

[2] Aderinwale, T. et al. (2022) "Real-time structure search and structure classification for alphafold protein models," *Communications Biology*, 5(1)

REVIEWERS' COMMENTS:

Reviewer #1 (Remarks to the Author):

The authors have addressed most of my concerns and added new experimental data. Thus, I believe that the manuscript improved substantially and should be published in its current form in Communications Biology.

Reviewer #3 (Remarks to the Author):

The reviewers' have addressed my concerns. Minor additions to the webserver are required:

a. I tried putting putting the query to the web-resource that recommended some waiting time. There should an email sent or easier option could an additional page where the person can read the results based on a "Job-ID". Authors may think of others ways in which if the tab is lost how can user retrieve the results??

b. Secondly the output is quite vast but different format downloaded options were missing. Can there be different files, like pdb IDs of proteins that are similar, expanded results as separate?

Lastly, due to lot of modifications in the last revision, many mistakes or overall flow needs to be corrected. The authors may be given some time to polish the paper better.

Reviewers' comments

Reviewer #3 (Remarks to the Author):

The reviewers' have addressed my concerns. Minor additions to the webservice are required:
a. I tried putting putting the query to the web-resource that recommended some waiting time. There should an email sent or easier option could an additional page where the person can read the results based on a "Job-ID". Authors may think of others ways in which if the tab is lost how can user retrieve the results??

Response:

We designed the service with simplicity in mind. The user has to perform a minimum set of actions to use MachaonWeb, retaining his privacy (no e-mail is required). The results of the query are cached temporarily (according to the total capacity of the system) that will be enough for the user to access the results without delay (provided that the job has been completed successfully), by performing an identical submission. We updated the "Instructions" section of the platform accordingly. In the next development iteration, the platform will store locally any request links in an overlapping list at a fixed position in the platform's interface that the user can view and manage, preserving the simplicity/privacy scheme.

b. Secondly the output is quite vast but different format downloaded options were missing. Can there be different files, like pdb IDs of proteins that are similar, expanded results as separate?

Response:

Machaon generates files that correspond to different phases of the analysis. The majority of the files are compatible with spreadsheet software like Excel where one can effortlessly retrieve any slice of the results. In the "candidates" folder, files that are suffixed with "-merged-enrich.csv" filename contain the least information (IDs and metric values), thus they are more manageable for simple needs such as the one described in comment b. There is already an extensive documentation about the outputs of the method in the GitHub repo of Machaon and we also added a brief mention about it in the instructions of the platform.

Lastly, due to lot of modifications in the last revision, many mistakes or overall flow needs to be corrected. The authors may be given some time to polish the paper better.

Response:

The submitted manuscript has been refined for the reader's convenience.